# INTERPRETABLE NEURAL ODES FOR GENE REGULATORY NETWORK DISCOVERY UNDER PERTURBATIONS

## ABSTRACT

Modern high-throughput biological datasets with thousands of perturbations provide the opportunity for large-scale discovery of causal graphs that represent the regulatory interactions between genes. Numerous methods have been proposed to infer a directed acyclic graph (DAG) corresponding to the underlying gene regulatory network (GRN) that captures causal gene relationships. However, existing models have restrictive assumptions (e.g. linearity, acyclicity), limited scalability, and/or fail to address the dynamic nature of biological processes such as cellular differentiation. We propose *PerturbODE*, a novel framework that incorporates biologically informative neural ordinary differential equations (neural ODEs) to model cell state trajectories under perturbations and derive the causal GRN from the neural ODE's parameters. We demonstrate PerturbODE's efficacy in trajectory prediction and GRN inference across simulated and real over-expression datasets.

## 1 INTRODUCTION

GRNs capture the complex regulatory interactions between genes that dictate cell function, development, and responses to environmental changes. High-throughput perturbation assays with single-cell RNA sequencing (scRNA-seq) readouts, such as Perturb-seq, enable precise measurement of gene expression changes across cell types resulting from genetic perturbations. However, inferring GRNs from scRNA-seq experiments remains challenging due to the problem's exponential search space. To overcome the inherent combinatorial complexity of GRN discovery, recent causal graphical modeling approaches relax the problem into a continuous, albeit non-convex, optimization program that learns a directed acyclic graph (DAG) corresponding to the underlying GRN (Zheng et al., 2018; Fang et al., 2023; Brouillard et al., 2020; Lopez et al., 2022).

While causal graphical models have predominantly focused on learning structure from gene knockdown-based perturbations, new interventional single-cell experiments offer insights into previously unexplored aspects of gene regulation at an unprecedented scale. In particular, the Transcription Factor (TF) Atlas applied single-cell resolution assays to systematically study the effects of overexpression of 1,836 TFs on cell differentiation, generating over 1.1 million cell profiles measured 7 days following TF perturbation (Joung et al., 2023). TFs, proteins that bind to the genome to regulate gene expression, play a crucial role in defining cell states. Their overexpression can induce significant changes in cell fate, directing the differentiation of stem cells into various cell types such as myocytes and neurons. Since gene regulation during differentiation is inherently dynamic, accurately capturing these dynamics is essential for models aiming to uncover the underlying regulatory network. Previous experiments in yeast and E. coli have demonstrated that gene regulatory dynamics can be effectively modeled by complex non-linear dynamical systems (Alon, 2006; Setty et al., 2003; Kalir and Alon, 2004). Their experimentally validated GRNs contain negative self-loops, which contradicts the assumption of graph acyclicity imposed by most structure learning methods.

Causal structure learning methods are limited in their ability to model the full complexity of interventional data generated by emerging single-cell assays. To address these limitations, we propose *PerturbODE*, a novel neural ODE-based framework that 1) implicitly encodes the GRN in its parameters, enabling simultaneous trajectory inference and GRN discovery, 2) maps cell states into a lower dimensional "gene module" space analogously to causal representation learning (CRL), 3) allows explicit input of which gene(s) were perturbed, a feature uncommon in CRL approaches, 4) can model cycles and non-linear gene interactions, and 5) incorporates novel diffusion model-

inspired regularization of the system's dynamics. Trained on the TF Atlas scRNA-seq data that captures the differentiation pathways of cells perturbed by over-expression of over a thousand TFs, PerturbODE enables scalable and interpretable discovery of the gene dependencies that drive cellular differentiation.

## 2 RELATED WORK

**Causal graph discovery from genetic perturbations**. Structure learning of causal graphs has recently been applied to Perturb-seq interventional experiments to infer an underlying GRN. The nodes in the encoded causal graph correspond to genes and the directed edges ideally correspond to direct causal regulatory relationships between genes. Since the number of possible DAGs grows exponentially with the number of nodes, classical causal graph discovery approaches are unable to scale beyond a modest number of genes (typically 50-200). NO-TEARS (Zheng et al., 2018) introduced a continuous optimization objective via the trace exponential acyclicity constraint, significantly simplifying the problem complexity and enabling gradient descent-based structure learning. Extensions have further improved scalability. NO-TEARS-LR (Fang et al., 2024) adds a low-rank assumption to NO-TEARS to efficiently infer large and dense DAGs. DCDI (Brouillard et al., 2020) extends the continuous optimization formulation to interventional data but can only scale up to 50 dimensions in their original implementation with the trace exponential acyclicity constraint. DCDFG (Lopez et al., 2022) addresses DCDI's limited scalability by employing a low-rank factor graph structure and spectral radius acyclicity constraint.

**Neural ODEs for cell trajectory inference and modeling gene regulation.** Differential equation-based models have long been the preferred framework for describing dynamical systems in biology due to their interpretability and flexibility in incorporating known properties of the system. Neural ODEs extend this framework by leveraging neural networks to learn the dynamics directly from data, making them particularly suited for modeling complex, high-dimensional systems without explicit formulations. Neural ODEs and their stochastic variants have been applied to trajectory inference, where the continuous development of cellular states is mapped over time. Some authors fit a discrete ODE specified by a gene regulatory function to temporal pairs of cells sampled from the optimal transport plan (Schiebinger et al., 2019). The gene regulatory function encodes information about the cell-autonomous regulatory networks. Jackson et al. (2023) parameterizes ODEs with recurrent neural networks (RNNs) to model dynamics before obtaining the coefficient of partial determination to represent the contribution of each TF.

**Causal Graph learning through stationary diffusion.** The recently proposed method BICYCLE (Rohbeck et al., 2024) parameterizes the GRN adjacency matrix as the linear drift of a stable Olstein-Uhlenbeck (OU) process, approximating the steady state distribution under each intervention induced by the OU process by solving the Lyapunov equation. Despite the novelty in methodologies, it can currently only handle tens of observed genes.

**Key Limitations**. Despite recent improvements to structure learning, causal GRN inference methods remain difficult to scale and are limited in their modeling capabilities. While they can learn some causal regulatory relationships from knockdown data, they lack the expressivity to capture how gene regulation affects cellular dynamics across time. Schiebinger et al. (2019) and Jackson et al. (2023) applied neural ODE-based methods for learning GRNs from a single perturbation or reprogramming trajectory, but provide no framework for leveraging datasets with multiple known genetic perturbations. PerturbODE combines ideas from causal structure learning and trajectory inference to provide a flexible and scalable framework that accurately captures cell dynamics and learns gene regulation from thousands of perturbations.

## 3 METHODS

Let $\mathcal{I} = \{I_0, I_1, \ldots, I_k\}$ represent a collection of $k + 1$ intervention regimes, with $I_0$ denoting the control regime (no intervention). The training dataset $\mathcal{D} = \{Y_i\}_{i=0}^k$ represents gene expression data, where $Y_i \in \mathbb{R}^{n_i \times d}$ corresponds to the $d$-dimensional gene expression measurements for $n_i$ cells under intervention regime $I_i$. Since $Y_0$ corresponds to unperturbed samples, it is used as the initial state from which we integrate our neural ODE function $f_i$ to predict the perturbation effect under a given intervention.

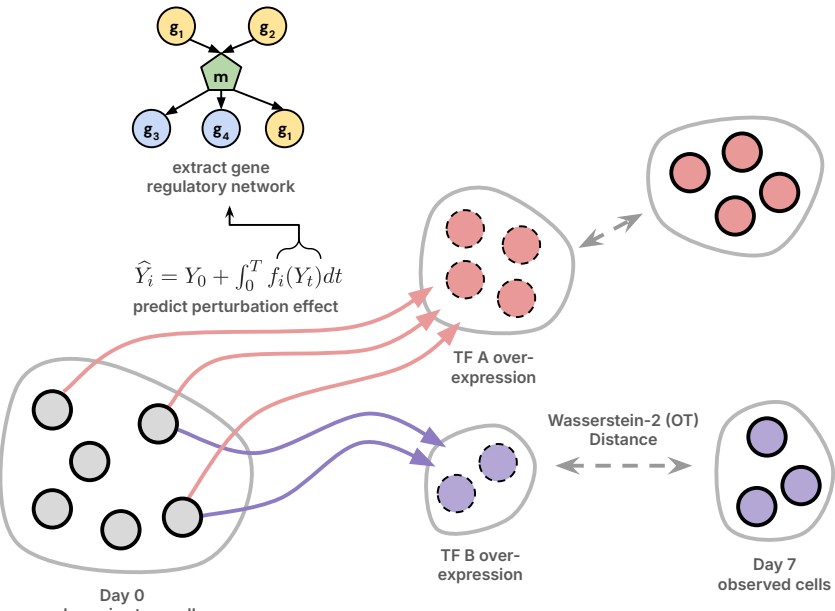

Figure 1: PerturbODE models the effect of a TF perturbation on stem cell differentiation by integrating the learned neural ODE function $f$ from the initial distribution of stem cell gene expression $Y_0$ under intervention $i$. The predicted gene expression values $\widehat{Y}_i$ are then compared to the observed differentiated expression values using the Wasserstein distance. From the parameters of the neural ODE function, we extract an underlying GRN that represents the regulatory relationships through gene modules. The GRN gene modules can capture cycles as well as both positive and directed edges.

### 3.1 NEURAL ODE FORMULATION FOR OVER-EXPRESSION WITH IMPERFECT INTERVENTION

For each cell subjected to an intervention $I_i \in \mathcal{I} \setminus \{I_0\}$, which specifies a set of over-expressed genes, its cellular dynamics is described by the ODE,

$$\frac{\partial Y_i}{\partial t} = f_i(Y_t) = A\sigma(\alpha \circ (BY_t - \beta)) + \sum_{j \in I_i} s_j \cdot \delta_j - WY_t, \tag{1}$$

where $Y_t \in \mathbb{R}^d$ represents the expression vector of a given cell at time $t$ for $d$ genes.

This system encapsulates the interaction between genes through a Multi-layer Perceptron (MLP) with a single hidden layer. Each neuron in the hidden layer is analogous to a gene module, similar to Segal et al. (2005). Such regulatory structure is known in the biology, such as production of flagella in E. coli (Macnab, 2003; Alon, 2006). For more details, appendix 6.10 illustrates the representation of the regulatory circuit of E. coli's flagella as a two-layer MLP.

The matrix $B \in \mathbb{R}^{l \times d}$ represents a linear transformation from the $d$-dimensional gene expression $Y_t$ to a lower $l$-dimensional latent ("module") space ($l \ll d$). $B_{jm}$ is the signed effect of $j$-th gene's expression on the the $m$-th module.

The gene module signals are then non-linearly transformed after shift and scaling to give module activations. The non-linear activation function $\sigma(\cdot) : \mathbb{R}^l \to \mathbb{R}^l$ models the activation of the gene modules, with the logistic sigmoid function used as the default choice for $\sigma(\cdot)$. This activation function is chosen because of its relationship to the Hill function, which is well-studied and biophysically motivated for representing the effect of TF concentration on target gene transcription rate (Alon, 2006). The vector $\beta \in \mathbb{R}^l$ is a strictly positive bias that shifts the activation threshold of the function $\sigma$. The vector $\alpha \in \mathbb{R}^l$ is a scaling factor that modulates the rate of activation through a Hadamard (i.e., elementwise) product ( $\circ$ ) with the gene modules.

The module activations regulate downstream genes by combining linearly with those from other modules. The matrix $A \in \mathbb{R}^{d \times l}$ maps the $l$-dimensional latent vector back to the $d$-dimensional gene expression space. $A_{mj}$ represents the influence of the $m$-th module on the $j$-th gene.

Most importantly, the interaction between genes mediated by modules encodes the GRN matrix, which we compute as $\mathbf{W} = A \operatorname{diag}(\alpha \circ \mathbf{1}_N) B$. Conveniently, working with the lower-dimensional module space reduces our task from learning the full gene-to-gene matrix of size $d \times d$ (i.e., $d^2$ parameters) to learning two factorized graphs of size $d \times l$ (i.e. $2dl$ parameters).

The matrix $W \in \mathbb{R}^{d \times d}$ is diagonal with strictly positive entries, such that $W_{ii} > 0$ is the decay rate for gene $i$. The decay component $-WY_t$ represents cellular RNA levels decreasing over time due to molecular decay and concentration dilution as the cells grow and divide. Decay is biologically well-motivated and encourages stability in the ODE system to prevent extreme expression by creating a trapping region.

Interventions on the system are captured by shifts, where $\delta_j = \mathbf{e}_j \in \mathbb{R}^d$ is a standard basis vector corresponding to the induced over-expression of gene $j$. Also, $s = (s_1, s_2, ..., s_d)^\top$ scales the strength of intervention on each gene. We generally fix $s$ to be a vector of ones. If needed, $s$ can become a model parameter updated during training. The vector encodes a 1 in the $j^{th}$ entry and 0 in all other entries, enabling variable dynamics between cells with over-expression of different TFs. Importantly, $\delta_j$ is the only parameter manually set according to the perturbed genes to reflect each intervention, while all other model parameters ($A$, $B$, $W$, $\alpha$, $\beta$, and $s$ if needed) are shared across all interventions.

### 3.2 NEURAL ODE FORMULATION WITH PERFECT INTERVENTION

PerturbODE is also capable of modeling perfect intervention. Gene knockout or over-expression (CRISPR-a) under perfect intervention is modeled by removing the intervened genes' dependencies on parent nodes. In a system subject to a set $S$ of perfect interventions, where $S$ contains the indices of the intervened genes, the corresponding ODE is,

$$\frac{\partial Y_i}{\partial t} = MA\sigma(\alpha \circ (BY_t - \beta)) + \sum_{j \in I_i} s_j \cdot \delta_j - WY_t \tag{2}$$

where $M = \mathbf{I} - \sum_{j \in S} \operatorname{diag}(\delta_j)$ is a masking matrix that removes the effect of other genes on the perturbed gene(s). For over-expression $s_j > 0$ for all $j$, whereas for knockout we set $s_j = 0$ for all $j$.

### 3.3 MAPPING DYNAMICS TO TARGETS USING OPTIMAL TRANSPORT

$f_i(Y_t)$ is learned by mapping the initial gene expression state $Y_0$ to the observed target state $Y_i$ by intervening genes specified by $I_i$. We compute our target predictions $\widehat{Y}_i$ by solving the ODE integration with the initial state $Y_0$,

$$\widehat{Y}_i = \phi_T^i(Y_0) = Y_0 + \int_0^T f_i(Y_t)dt \tag{3}$$

where $\phi_T^i(Y_0)$ is the flow map of the ODE under intervention $I_i$ mapping initial condition $Y_0$ to its position at time $T$ through the numerical solution to the ODE for this initial value problem.

Given the lack of one-to-one correspondence between samples (cells) in the initial distribution and the samples in the target distributions, we assess the quality of our predictions by measuring the Wasserstein-2 distance between observed targets $Y_i$ and predicted targets $\widehat{Y}_i$, i.e.

$$W_2(X, \widehat{X}) = \left( \min_{\Gamma \sim \Pi(X, \widehat{X})} \sum_{x,y} \|X_x - \widehat{X}_y\|_2^2 \Gamma_{xy} \right)^{1/2}, \tag{4}$$

where $\Pi$ represents the set of all optimal transport plans between each sample from data distributions $X$ and $\widehat{X}$, and $\Gamma$ represents the minimal-cost transport plan used to measure the dissimilarity between $X$ and $\widehat{X}$. The total loss function is defined as the average $W_2$ between $\widehat{Y}_i$ and $Y_i$ for all perturbed genes in addition to the $L_1$ norm of $B$ to encourage sparsity,

$$\mathcal{L}_i^\theta = W_2(Y_i, \widehat{Y}_i) + \lambda |B|. \tag{5}$$

During training, for each intervention $I_i$, we push the control samples $Y_0$ through the map $\phi_T^i$ to obtain the predicted targets $\widehat{Y}_i$. We backpropagate through the loss and ODE solver to obtain gradients for all parameters. $L_1$ penalty is enforced only on $B$ because the network motif (pattern) of multiple input feed-forward loop is significantly more rare than that of multiple output feed-forward loop in known GRNs of yeast and E. coli (Kashtan et al., 2004).

During each epoch, PerturbODE iterates through all intervention regimes in $\mathcal{I}$. Further details on data splitting and loss convergence can be found in Appendix 6.7.

### 3.4 DIFFUSION-BASED REGULARIZATION OF NEURAL DYNAMICS

PerturbODE can optionally augment the primary training objective by using diffused target samples as alternative initial states. This additional regularization encodes our prior expectation that the final cell states should be locally stable, helping to form a local contraction map that implies a locally stable fixed point, as ensured by the Contraction Mapping Theorem (Hunter and Nachtergaele, 2000). Further, the stable fixed points establish the theoretical equivalence between PerturbODE and a deterministic structural causal model (SCM), endowing it with its causal mechanism (Mooij et al., 2013; Schölkopf et al., 2021).

The augmentation involves diffusing $Y_i$ using Brownian motion with a time step $\Delta t$ to generate diffused targets $\tilde{Y}_i$. Across a reduced time span $t \leq T$, $\tilde{Y}_i$ is pushed forward through $\phi_t^i$ to obtain the predicted targets $\widehat{Y}_i{}'$, and we backpropagate against the augmented loss $\tilde{\mathcal{L}}_i = W_2(\widehat{Y}_i{}', Y_i) + \lambda |B|$. During training, we alternate between using control samples $Y_0$ and diffused targets $\tilde{Y}_i$ for each intervention. Information on the exact training hyperparameters can be found in Appendix 6.2.2.

## 4 RESULTS

We compare PerturbODE to the causal graph discovery methods DCDFG (Lopez et al., 2022), DCDI (Brouillard et al., 2020), NO-TEARS (Zheng et al., 2018), and NO-TEARS-LR (Fang et al., 2023) through extensive experiments on both simulated and large-scale perturbational scRNA-seq datasets. These methods are good comparisons since, similar to PerturbODE, they also embed GRNs as matrices either in neural networks or directly in linear models.

PerturbODE not only infers cycles but also detects both positive and negative edges, whereas the DCDI and DCDFG only identify edge existence under the DAG constraint. To enable benchmarking, the ground truth GRNs used in simulated data are DAGs with positive edges, and we validate solely against positive edges in the reference GRNs in the scRNA-seq dataset. As ground truth negative edges are unavailable for evaluation, we classify any negative edge inferred by PerturbODE as the absence of an edge. This setup gives PerturbODE a more difficult task in predicting the correct edge sign and prevents it from leveraging its full range of capabilities. Therefore, we provide further downstream analysis on PerturbODE's model parameters when trained on real datasets, showcasing its strengths in uncovering network structures through its biologically faithful and interpretable modeling approach.

### 4.1 GRN INFERENCE ON SERGIO SIMULATED DATASETS

SERGIO (Dibaeinia and Sinha, 2020) simulates single-cell gene expression data by modeling gene regulation of each gene by multiple TFs according to a user-provided DAG representing the GRN. SERGIO can simulate any number of cell types in steady state or cells differentiating to multiple fates. The simulator samples single-cell gene expression data through a stochastic differential equation (SDE) initialized at the expected steady state.

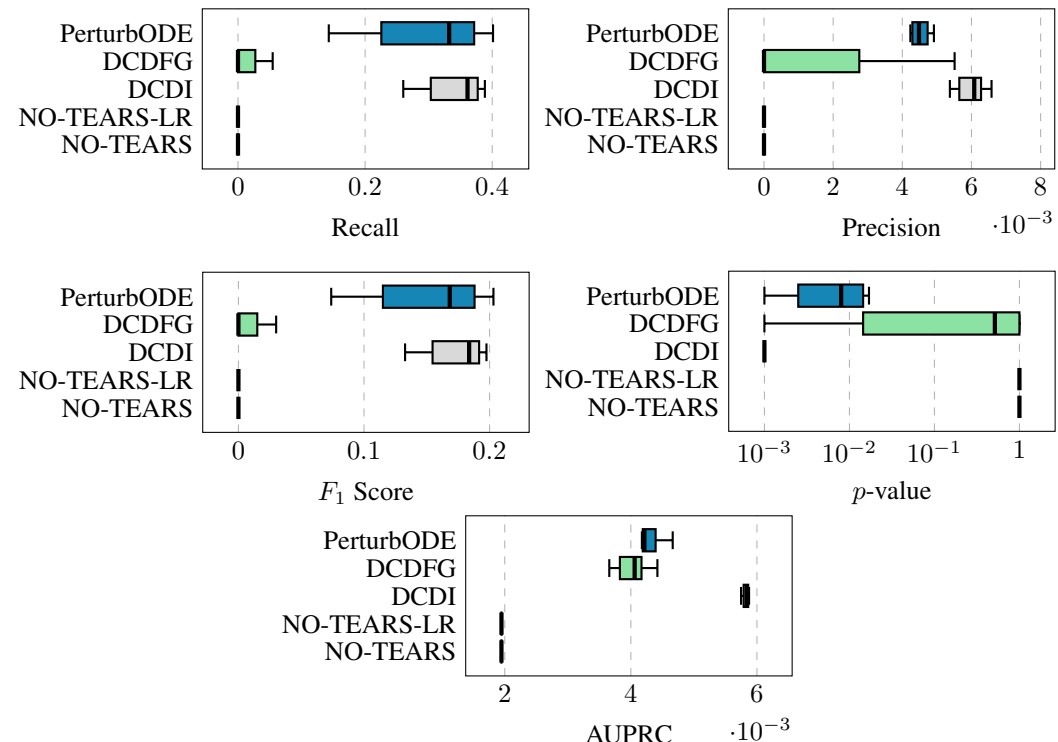

Figure 2: Performance metrics on SERGIO-simulated data of a known yeast GRN (400 genes), assuming perfect intervention over-expression (CRISPR-a).

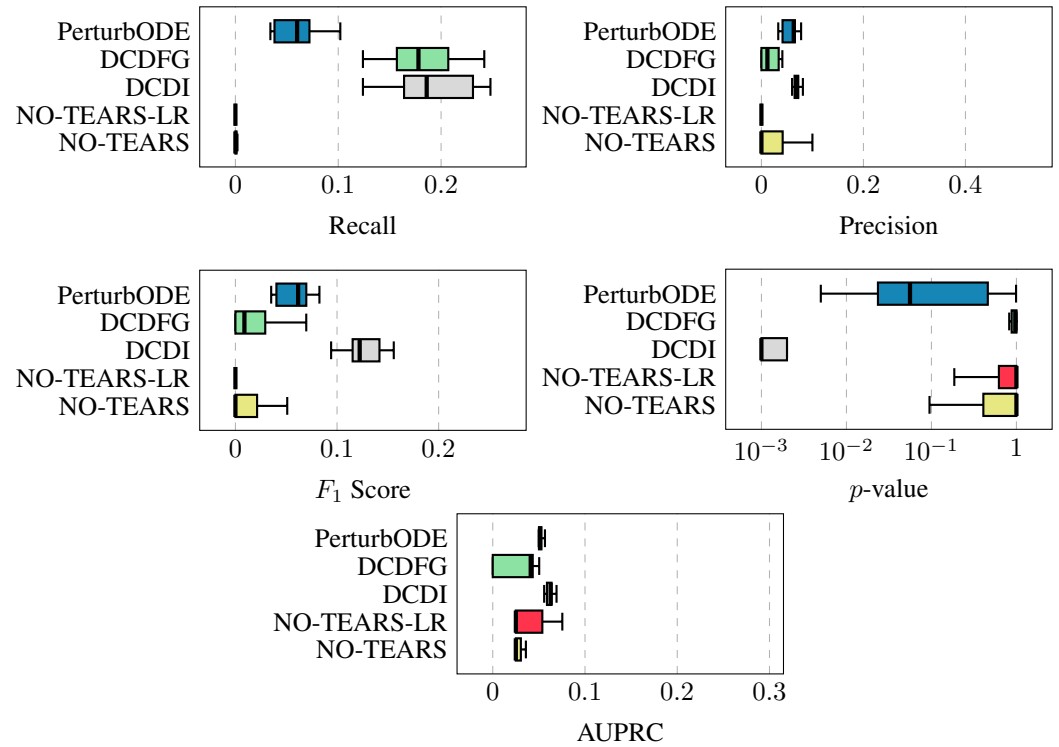

Figure 3: Performance metrics on SERGIO-simulated data of 10 random acyclic GRNs (100 genes), assuming perfect intervention over-expression (CRISPR-a).

We extend SERGIO to simulate data with over-expression genetic perturbations. We implement interventions by masking the protein production induced by TF interactions (analogously to $M$ in Equation 2) of the intervened genes and adding a scalar over-expression to the intervened gene's transcription rate. We select an experimentally validated GRN identified for yeast cells with dimension $400$ as the input to SERGIO for simulation. The GRN is pruned to fit the required DAG constraints. The output synthetic dataset from SERGIO consists of 10,100 cells generated from 100 intervention schemes each targeting 5 genes and one non-intervention scheme. Each regime contains 100 cells. Similarly, ten random DAGs with dimension 100 are generated with data simulated in the same manner.

For evaluation, we threshold the weights of the output GRNs to obtain classification metrics based on edge detection (details in Appendix 6.2.1). The recall, and consequently the F1 score, can be strongly influenced by the number of edges returned by the model. If the model consistently predicts full graphs, the recall may be artificially inflated. Therefore, we evaluate the AUPRC across models. DCDI achieves a higher AUPRC than PerturbODE, and PerturbODE outperforms all other models. To further address the discrepancies between graph sparsity and predictive performance, we employed random graphs to generate an empirical null for each test statistic for random graphs with the same edge density. We compare the precision-recall test statistics of the predicted GRN against those from $10,000$ Erdős-Rényi random networks, yielding empirical $p$-values (for details, see 6.3). Note the Structural Hamming Distance (SHD) would strongly favor predictions of empty graphs since the ground truth GRNs are extremely sparse and high dimensional, so we decided not to use SHD for evaluating model performance.

There is considerable variation in recall scores for PerturbODE especially in the simulated yeast dataset. This is likely due to the high sparsity in the ground truth GRN, which leads to weak signals in the simulated dataset. This results in false negatives. Further, $L_1$ penalty is enforced on the individual matrix. As multiplication of sparse matrices is not always sparse, the number of predicted edges tend to fluctuate. Denser predictions would have higher recall scores.

PerturbODE demonstrates significantly higher precision, recall, F1, and AUPRC scores compared to DCDFG, NO-TEARS, and NO-TEARS-LR, while performing comparably to DCDI in these metrics (Fig. 3, Fig. 2). DCDI is the state-of-the-art method that outperforms PerturbODE in lower dimensional simulated datasets ($100 - 400$ genes), but it lacks scalability. In fact, for dimensions greater than $400$, DCDI simply fails to execute, even with the more computationally feasible spectral radius acyclicity constraint. Details of the performance across all models with varying numbers of modules are provided in 6.5.4. PerturbODE's main contribution is its ability to train on real datasets with thousands of genes, while maintaining competitive predicative performance.

## 4.2 GRN INFERENCE ON THE TF ATLAS

We trained PerturbODE on the TF Atlas to evaluate its performance on large-scale real experimental datasets. The TF Atlas over-expresses TFs and uses scRNA-seq to measure cell states after 7 days of perturbation (Joung et al., 2023). As this dataset maps the interventional effects of TF over-expression, PerturbODE's inferred GRNs can uncover TF-to-TF interactions and higher-level network structure through TF modules.

In this setup, we used the control samples (mCherry) as the initial gene expression state for solving the neural ODE, while the final gene expression states correspond to cells with TF over-expressions that induced differentiation after 7 days. We evaluate the model's performance using three well-studied and experimentally validated human GRNs derived from extensive RNA-seq and ATAC-seq measurements. See Appendix 6.8 for further details on the GRNs used for evaluation. Notably, the ground truth GRNs only contain positive directed edges, restricting our evaluation to true positives and false negatives for benchmarking GRN edge detection. Consequently, we compute $p$-value and total recall score based on predictions of directed edges across all three GRNs. In addition, Figure 9 in Appendix shows recall scores across models in various sparsity levels.

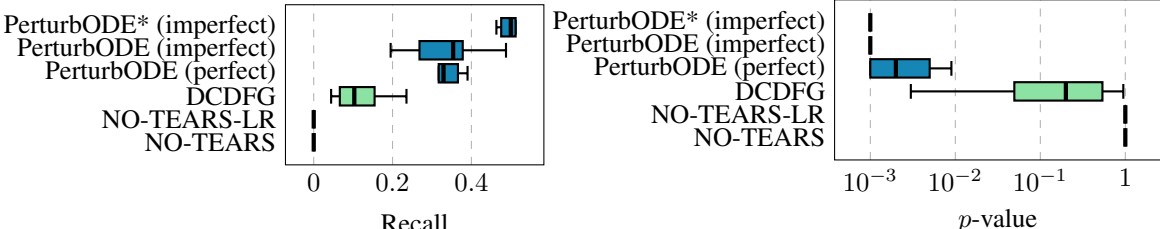

Figure 4: GRN inference performance on TF Atlas dataset (817 genes)

We compare to DCDFG, NO-TEARS, and NO-TEARS-LR by training on the union of the top 500 highly variable genes and experimentally intervened genes that are differentially expressed (817 genes in total). DCDI cannot handle datasets of this scale, making it unsuitable for comparison. PerturbODE's GRN estimation under both perfect and imperfect intervention models are evaluated. PerturbODE significantly outperforms DCDFG, NO-TEARS, and NO-TEARS-LR in recall with more significant $p$-value (Fig. 4). PerturbODE* denotes the version with tunable over-expression strength for each gene. PerturbODE* with imperfect intervention is the best performing model in terms of recall scores and p-values in this dataset.

### 4.2.1 PREDICTION OF HELDOUT INTERVENTIONS

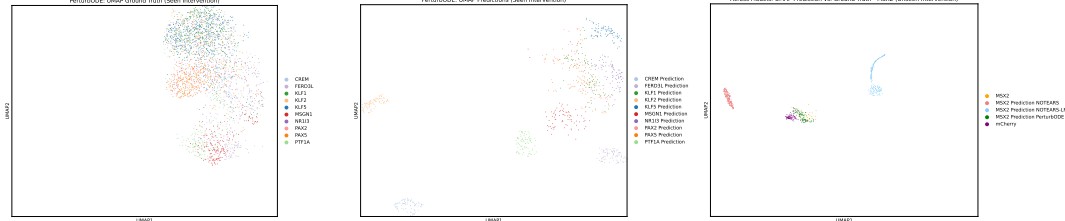

Figure 5: Visualization of the cell embeddings for the trajectory predictions of 10 transcription factors (TFs) in the training set and a held-out TF—MSX2—in the test set using UMAP1 and UMAP2. Each point represents a cell's embedding in the reduced-dimensional space.

Predicting the effects of unseen, i.e., heldout, interventions is a particularly challenging task. Here we randomly select ten overexpressed TFs to be held out simultaneously during training. Note that their expression levels are observed, but their perturbations are not trained on. For this task, we only compare PerturbODE with linear SCMs (NO-TEARS and NO-TEARS-LR). DCDFG cannot sample cells given a learned GRN, and DCDI lacks scalability for large datasets. For the linear SCMs, over-expression is implemented as imperfect shift intervention by adding a bias to the mean of the distribution modeling the intervened nodes (for details, see Appendix 6.4).

We evaluate the predictive performance through Pearson correlation, $W_2$ distance between the distributions, and manual inspection via low dimensional embeddings. Pearson correlation is computed between the average predicted gene expression and the average gene expression of experimentally perturbed cells, while $W_2$ distance is calculated between the full distributions of predicted and observed gene expressions.

Table 1: Predictive performance on 10 held-out interventions in TF-Atlas

| METHOD (MEDIAN) | PERTURBODE | NO-TEARS | NO-TEARS-LR |
|---|---|---|---|
| $W_2$ DISTANCE | $84.02 \pm 183.88$ | $396.87 \pm 232.11$ | $105.14 \pm 1.2$ |
| PEARSON CORRELATION | $0.67 \pm 0.14$ | $-0.03 \pm 0.02$ | $0.03 \pm 0.01$ |

PerturbODE outperforms the other methods in terms of median correlation $W_2$ distance with the held-out interventions (Table. 1). To give some context to the scale of $W_2$, we note that before model training, the predicted target distributions have an average $W_2$ distance of over 2000 from the ground truth distributions. The model's training and validation loss curves can be found in Appendix 6.7.

When we visualize our predictions compared to the linear SCMs across held-out TFs through UMAP, we show that PerturbODE's predictions are much closer to the observed distributions (Figure 5 and Appendix 6.6.1). We conclude that PerturbODE is learning the TF-TF regulatory relationships sufficiently well to generalize its predictions of over-expression dynamics to unseen TFs.

### 4.2.2 NEGATIVE AUTOREGULATION IN PERTURBODE INFERRED GRNS

PerturbODE's unique ability to learn cyclic GRNs sets it apart from other causal methods that assume acyclicity. Cycles, especially negative autoregulation, are known to be a prevalent network motif in gene regulation. Negative autoregulation accelerates response times by enabling quicker adjustments to input signals and enhances robustness by stabilizing gene expression levels against fluctuations in production rates (Alon, 2006). Approximately 40% of known E. coli TFs exhibit negative feedback regulation (Rosenfeld et al., 2002).

When trained on the TF Atlas, PerturbODE naturally incorporate this network motif (pattern) without the need for explicit priors. The model predicts that 26.4% of modeled genes are subject to negative autoregulation, which aligns with the expected prevalence of the motif according to prior studies. To assess the statistical significance, we numerically compute the frequency of negative self-loops in random graphs with the same graph density, yielding a highly significant $p$-value of less than 0.001. The result underscores both the statistical significance and biological realism of PerturbODE's predictions. By inferring the GRN from interventional dynamics, PerturbODE could learn network structures that can not be captured by strictly acyclic approaches.

### 4.2.3 ANALYSIS OF INFERRED GENE MODULES

PerturbODE's framework enables direct interpretation of the inferred gene modules, which encapsulate multiple gene to gene interactions. These interactions are extracted from the $A$ and $B$ matrices (Eq. 1), where the entries in $B$ represent directed edges from upstream genes to gene modules, and the entries in $A$ map the modules to downstream genes.

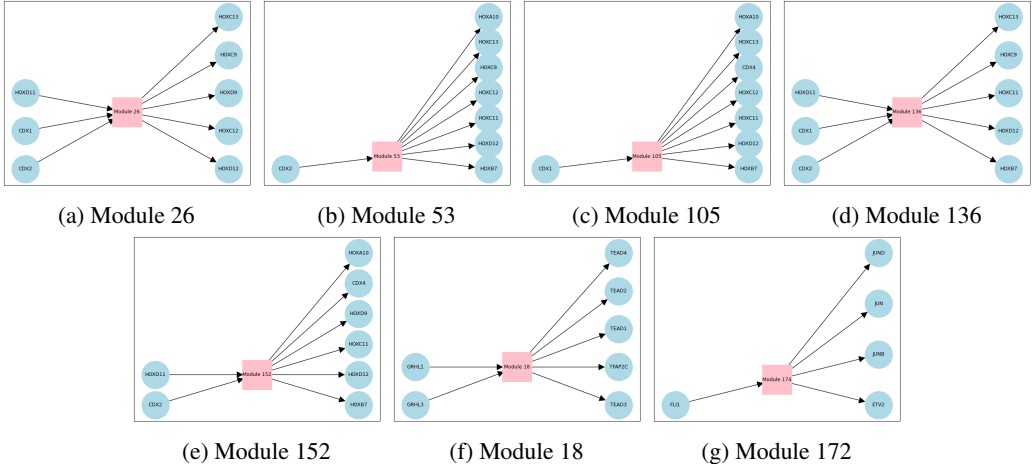

  (a) Module 26         (b) Module 53         (c) Module 105        (d) Module 136

      (e) Module 152         (f) Module 18        (g) Module 172

Figure 6: Modules identified by PerturbODE that align with established regulatory relationships.

To highlight the advantages of PerturbODE's interpretability, we analyze the 200 inferred latent gene modules obtained from training on the TF Atlas dataset and visualize seven modules that correspond to directed edges found in experimentally validated GRNs (Fig. 6). The modules with the highest test statistic scores (Section 6.3) are shown. The modules in (a) - (e) encapsulate the GRN responsible for specification of the anterior-posterior axis in development (Neijts et al., 2017). (f) and (g) successfully

capture known GRNs responsible for inducing trophoblasts and vascular endothelial cells respectively (Krendl et al., 2017; Dejana et al., 2007). Additionally, we compare to Erdős-Rényi random matrices for all the inferred modules, resulting in average p-values less than 0.001 (Appendix 6.3). These visualizations demonstrate that the modules recover the appropriate gene network structure, clustering genes from the same GRN and accurately inferring edges between them.

Figure 7 presents a clustered heatmap of the enrichment analysis for modules in (a) to (g). (More details can be found in Appendix 6.12.) Modules 172 and 136 are enriched in pathways related to vascular endothelial cells as well those related to anterior-posterior axis specification. Meanwhile, modules 172, 136, 18, and 53 show clear enrichment in anterior-posterior axis specification with module 53 having the most significant enrichment. Module 18 demonstrates significant enrichment in pathways related to angiogenesis and response to fluid stress.

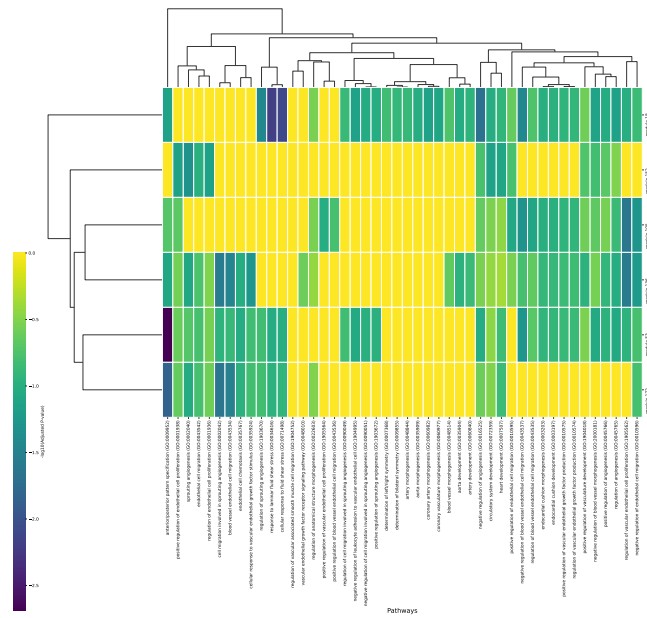

Figure 7: Gene enrichment clustered heatmap (average linkage) for selected modules.

## 5 CONCLUSION

PerturbODE's main contribution is a highly scalable and biologically realistic approach to discover gene regulatory network with thousands of genes from perturbation data. Given that dynamical systems are well-established for modeling gene regulation and have seen substantial success for single trajectory inference, PerturbODE presents a compelling alternative to traditional SCM methods. Using a two-layer neural network with sigmoid activation, we can achieve a close approximation of the actual cellular regulatory processes. Our framework ensures both strong predictive performance and biological interpretability of the learned parameters. For GRN inference, PerturbODE outperforms existing scalable methods on SERGIO-simulated datasets and large-scale single-cell experiments, while performing competitively against state-of-the-art methods, such as DCDI. It is also capable of predicting cellular responses to unseen perturbations. Future work will focus on understanding the conditions under which identifiability is assured to further solidify PerturbODE's theoretical foundations.

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

# 6 Appendix

## 6.1 Preprocessing

The scRNA-seq gene expression matrix is normalized per cell by $10^4$ and $\log(1 + X)$ transformed. The total gene expression vector comprises RNA counts for $N$ genes consisting of all the TF over-expression genes $j$ and the top $k = 817$ variable genes.

For each TF gene $j$, we perform a Mann-Whitney U test on differential gene expression of TF $j$ between the unperturbed control samples in $X_0$ and over-expressed samples in $X_j$ consisting of $n_j$ cells. The returned p-value $p_j$ from the U test determines whether over-expression of the targeted TF gene $j$ is sufficiently induced in the experiments. The dataset is then filtered based on the criteria $\mathcal{D} = \{X_j \mid p_j < 0.1 \text{ and } n_j \geq 10, \ \forall j \in \{1, 2, \ldots, M\}\}$.

Over-expression distributions of the genes encoding the GRNs of interest are added to the training and validation dataset. In addition, when training for GRN inference only without trajectory prediction, distributions of TF over-expression encoded by the marker genes of the cell types or the developmental role targeted by the genes in the GRNs are included in the joint train, test, and validation dataset.

We design a train-test split based on TF over-expression genes to select $\mathcal{D}_{\text{train,val}}$ and $\mathcal{D}_{\text{test}}$. For each $X_j \in \mathcal{D}_{\text{train,val}}$ where $n_j \geq 100$, we apply a 80% to 20% training-validation split of the over-expression samples. If $n_j < 100$, we would use all the samples in $X_j$ for $D_{\text{train}}$ due to an insufficient number of training samples.

Furthermore, we apply the **log1p** transformation to prevent negative predictions of gene expression and mitigate length biases in expression counts (Gorin and Pachter, 2023). This transformation results in a substantial improvement in model performance.

## 6.2 Model Specifications

PerturbODE utilizes adaptive Runge-Kutta of order 5 of Dormand-Prince-Shampine which provides an exceptionally high order of accuracy and leverages its adaptive step size for efficient ODE solving. The adaptive step size also detects and handles a wide range of stiff ODEs. Differentiable numerical solution is computed via the adjoint method implemented in PyTorch by Chen (2021), available at `https://github.com/rtqichen/torchdiffeq`. The Sinkhorn-based $W_2$ distance is differentiable through the *GeomLoss* implementation in PyTorch (Feydy et al., 2019).

For the baseline methods, the authors of DCDFG have implemented DCDI, DCDFG, NO-TEARS, and NO-TEARS-LR in the repository Lopez (2024), available at `https://github.com/Genentech/dcdfg`.

### 6.2.1 Thresholds

We apply a threshold $\epsilon$ to the GRN matrix $\mathbf{W}$, where any edge with a weight below $\epsilon$ is set to 0 and any edge whose weight exceeds $\epsilon$ is set to 1.

PerturbODE's $\epsilon$ threshold is determined using the formula $\epsilon = c \cdot \sigma$, where $\sigma$ represents the standard deviation of the inferred GRN matrix $\mathbf{W}$ across all entries, and $c$ is a positive scalar. For SERGIO simulated data with 400 genes, $c = 0.1$, while for SERGIO simulated data with 100 genes and TF Atlas, $c = 0.01$. $c$ is chosen so that the PerturbODE predicts a reasonable number of edges (no more than 30% of possible edges). A lower threshold is chosen for the clarity of presentation by getting similar number of edges as DCDI.

As recommended by their authors, DCDFG determines the threshold $\epsilon$ through binary search, using depth of 20 evaluations of an exact acyclicity test to find the largest possible DAG for each method. NO-TEARS and NO-TEARS-LR's $\epsilon$ are chosen to be 0.3 while DCDI's is set to 0.5 as recommended by the respective authors. For DCDI, NO-TEARS and NO-TEARS-LR different thresholdings such as binary search are attempted without meaningful change to the result. Different fixed values for $\epsilon$ were also experimented for DCDFG without improvements.

### 6.2.2 HYPERPARAMETERS

Spectral radius is used as the DAG constraint for DCDI, DCDFG, NO-TEARS, and NO-TEARS-LR. Notably, NO-TEARS and DCDI fail to run at dimensions higher than tens of variables with the trace exponential constraint, making experiments using the original DCDI implementation infeasible. As recommended by the authors, we set the optimizer learning rate to 0.001 and the regularization coefficient to 0.1.

The number of modules is optimally set to 10 for NO-TEARS-LR and DCDFG. For PerturbODE, we set the number of modules to 100 for simulated data and 200 for TF Atlas. Details on performances across different number of modules in all models can be found in Figure 10.

As the number of modules increases, the model becomes closer to approximating the full graph. On the TF Atlas dataset, we demonstrate that the validation loss for PerturbODE decreases as the number of modules increases, plateauing after reaching 200 modules when training on TF Atlas (Fig. 8).

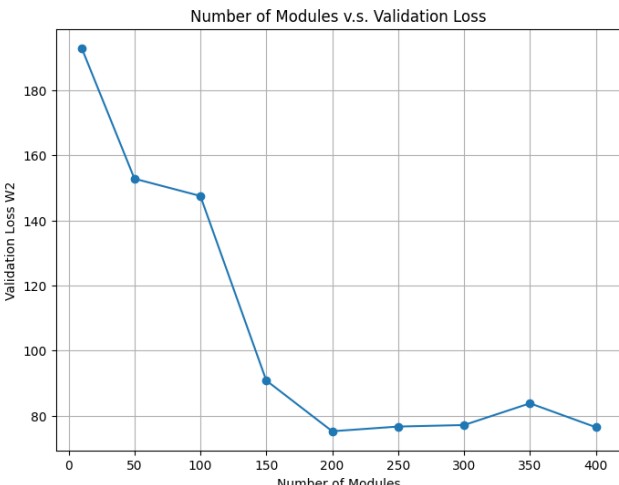

Figure 8: PerturbODE: number of modules v.s. validation loss in TF Atlas

On a separate note, PerturbODE uses 50 time steps for both diffused and non-diffused training when solving the ODE numerically. For diffused training, the time step duration $t$ is set to 0.1, while for non-diffused training, it is set to 25. The lasso regularization coefficient, $\lambda$, is set to 0.001. When computing the $W_2$ distance through Sinkhorn's algorithm, the coefficient for entropic regularization is set to 0.05. $\Delta t$ for the Brownian motion used to generate diffused data is set to 0.3.

### 6.3 COMPARISON TO ERDŐS-RÉNYI RANDOM GRAPHS

We generate $10,000$ random graphs with the same density as our inferred GRN to numerically simulate the test statistics under Erdős-Rényi random matrices. The p-value is calculated using the equation,

$$p\text{-value} = \frac{1 + \#\{\tau^* \geq \tau\}}{1 + \Pi} \quad (6)$$

where $\tau$ is the test statistic, $\Pi$ indicates the total number of random graphs, and $\tau^*$ denotes the test statistics computed from each graph. The p-value quantifies how often a test statistic is observed (or a more extreme one) purely by chance.

When evaluating SERGIO simulated data, the test statistics used is the F1 score, whereas recall score is used for TF Atlas due to availability of only positive benchmark edges. To identify gene modules, we use test statistics based on the count of incoming edges to the module and outgoing edges from the module that are consistent with known regulatory relationships. Further, to identify the network motif of negative auto-regulation, test statistics is the number of negative self-loops.

## 6.4 SAMPLING FROM LINEAR SCMs FOR TF ATLAS

For a learned GRN represented by $\mathbf{W}$ (ensured to be a DAG, or thresholded to enforce acyclicity), we sample from linear structural causal models (SCMs) using the following procedure. First, for each parent gene $i$ (master regulator) in the GRN, if not over-expressed, its expression level $X_i$ is sampled from a normal distribution, $X_i \sim \mathcal{N}(\mu, \sigma)$, where $\mu$ and $\sigma$ represent the mean and standard deviation of gene expression levels across all genes and cells in the TF Atlas, respectively. If $X_i$ is over-expressed, it is instead sampled from $X_i \sim \mathcal{N}(\mu_\gamma, \sigma_\gamma)$ where $\mu_\gamma$ and $\sigma_\gamma$ are the mean and standard deviation of gene expression levels in over-expression genes across all over-expressed cells.

Downstream genes are realized in Equation 7:

$$
\begin{aligned}
X_i &= \sum_{X_j \in \mathrm{pa}(X_i, \mathbf{W})} \mathbf{W}_{j,i} X_j && \text{if } X_i \text{ is not over-expressed,} \\
X_i &= \sum_{X_j \in \mathrm{pa}(X_i, \mathbf{W})} \mathbf{W}_{j,i} X_j + \gamma_i, \quad \gamma_i \sim \mathcal{N}(\mu_\gamma - \mu, \sigma_{\Delta\gamma}) && \text{if } X_i \text{ is over-expressed,}
\end{aligned}
\tag{7}
$$

where $\sigma_{\Delta\gamma}$ is the standard deviation of the differences between over-expressed genes and mean expression levels (average over genes) across all over-expressed cells. Further, $\mathrm{pa}(X_i, \mathbf{W})$ denotes all the parent genes (regulators) of gene $i$ in the GRN $\mathbf{W}$.

## 6.5 ADDITIONAL RESULTS

### 6.5.1 MEAN AND STANDARD DEVIATION OF RESULTS

| Method | Recall | | Precision | | AUPRC | | F1 | | p-value | |
|---|---|---|---|---|---|---|---|---|---|---|
| | Mean | Std | Mean | Std | Mean | Std | Mean | Std | Mean | Std |
| PerturbODE | 0.3191 | 0.0937 | 0.0046 | 0.0003 | 0.0044 | 0.0002 | 0.1618 | 0.0468 | 0.0212 | 0.0260 |
| DCDFG | 0.0315 | 0.0414 | 0.0026 | 0.0032 | 0.0041 | 0.0003 | 0.0170 | 0.0223 | 0.6058 | 0.4829 |
| NO-TEARS-lr | 0.0000 | 0.0000 | 0.0000 | 0.0000 | 0.0027 | 0.0015 | 0.0000 | 0.0000 | 1.0000 | 0.0000 |
| NO-TEARS | 0.0000 | 0.0000 | 0.0000 | 0.0000 | 0.0019 | 0.0000 | 0.0000 | 0.0000 | 1.0000 | 0.0000 |
| DCDI | 0.3499 | 0.0470 | 0.0061 | 0.0004 | 0.0059 | 0.0001 | 0.1780 | 0.0237 | 0.0010 | 0.0000 |

Table 2: Mean and standard deviation across models for yeast simulated by SERGIO

| Method | Recall | | Precision | | AUPRC | | F1 | | p-value | |
|---|---|---|---|---|---|---|---|---|---|---|
| | Mean | Std | Mean | Std | Mean | Std | Mean | Std | Mean | Std |
| DCDI | 0.3499 | 0.0470 | 0.0061 | 0.0004 | 0.0059 | 0.0001 | 0.1780 | 0.0237 | 0.0010 | 0.0000 |
| NO-TEARS-lr | 0.0000 | 0.0000 | 0.0000 | 0.0000 | 0.0027 | 0.0015 | 0.0000 | 0.0000 | 1.0000 | 0.0000 |
| DCDFG | 0.0315 | 0.0414 | 0.0026 | 0.0032 | 0.0041 | 0.0003 | 0.0170 | 0.0223 | 0.6058 | 0.4829 |
| PerturbODE | 0.3191 | 0.0937 | 0.0046 | 0.0003 | 0.0044 | 0.0002 | 0.1618 | 0.0468 | 0.0212 | 0.0260 |
| NO-TEARS | 0.0000 | 0.0000 | 0.0000 | 0.0000 | 0.0019 | 0.0000 | 0.0000 | 0.0000 | 1.0000 | 0.0000 |

Table 3: Mean and standard deviation across models for random DAGs simulated by SERGIO

| Method | Recall | | p-value | |
|---|---|---|---|---|
| | Mean | Std | Mean | Std |
| NO-TEARS | 0.0000 | 0.0000 | 1.0000 | 0.0000 |
| NO-TEARS-lr | 0.0000 | 0.0000 | 1.0000 | 0.0000 |
| DCDFG | 0.1353 | 0.0692 | 0.4158 | 0.3692 |
| PerturbODE (imperfect interv) | 0.3659 | 0.0556 | 0.0042 | 0.0032 |
| PerturbODE* (imperfect interv) | 0.4976 | 0.0195 | 0.0010 | 0.0000 |
| PerturbODE (perfect interv) | 0.3561 | 0.0946 | 0.0236 | 0.0452 |

Table 4: Mean and standard deviation across models for TF Atlas

### 6.5.2 PREDICTION ON UNSEEN INTERVENTIONS (INDIVIDUAL TFS)

| TF Over-expression | PerturbODE | NO-TEARS-LR | NO-TEARS |
|---|---|---|---|
| ZNF69 | 85.3758 | 106.0157 | 164.8816 |
| SETDB1 | 261.9399 | 97.1853 | 157.8617 |
| POU2AF1 | 300.8073 | 105.4930 | 163.0949 |
| ZBTB37 | 69.4434 | 107.1228 | 165.9257 |
| IRF3 | 73.6372 | 111.1662 | 170.1261 |
| ID1 | 79.6410 | 109.7050 | 168.6616 |
| TEAD1 | 244.5535 | 106.0757 | 163.4510 |
| ASCL1 | 94.0845 | 134.7678 | 192.7295 |
| KCNIP4 | 82.6612 | 104.7195 | 163.7381 |
| MSX2 | 66.6919 | 103.6894 | 164.6299 |

Table 5: Test errors ($W_2$) for TF over-expressions across different models.

### 6.5.3 NUMBER OF EDGES PREDICTED

Table 6 presents the number of edges predicted by each model across different datasets using the recommended thresholds. NO-TEARS and NO-TEARS-LR often under-predict, frequently resulting in near-empty graphs. While PerturbODE tends to over-predict, its $p$-values in comparison to random Erdős-Rényi matrices remain statistically significant. Similarly, DCDFG and DCDI also over-predict, though to a lesser extent compared to PerturbODE. For simulated data, AUPRC (Figure 2, 3) is the more appropriate metric in evaluation of model performances. Nevertheless, for TF Atlas, there is no complete ground truth network but only known edges, leaving it impossible to compute AUPRC. Therefore, we plot the recall in different sparsity levels across models by varying the thresholds in Figure 9. PerturbODE outperforms all other methods when the sparsity above 2%.

Table 6: Average number of edges predicted by all methods across datasets

| METHOD | GROUND TRUTH | PERTURBODE | NO-TEARS | NO-TEARS-LR | DCDI | DCDFG |
|---|---|---|---|---|---|---|
| YEAST GRN ($dim = 400$) | 623 | 43655.0 | 0.0 | 0.0 | 24332.8 | 4293.8 |
| RANDOM DAGS ($dim = 100$) | 500 | 552.0 | 0.0 | 7.1 | 1423.7 | 215.1 |
| TF ATLAS ($dim = 817$) | N/A | 101404.2 | 438.0 | 76.0 | N/A | 72884.0 |

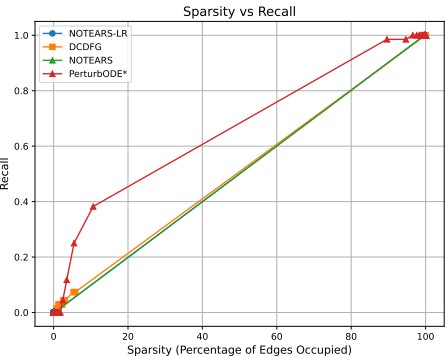

Figure 9: Mean and standard deviation across models for TF Atlas (817 genes)

### 6.5.4 GRN INFERENCE RESULTS WITH DIFFERENT NUMBER OF MODULES

PerturbODE and NO-TEARS-LR maintain consistent performance across different numbers of modules, while DCDFG achieves its best results with 10 modules. Figures 10 and 11 illustrate the performance of all models across varying module numbers in the SERGIO and TF Atlas datasets.

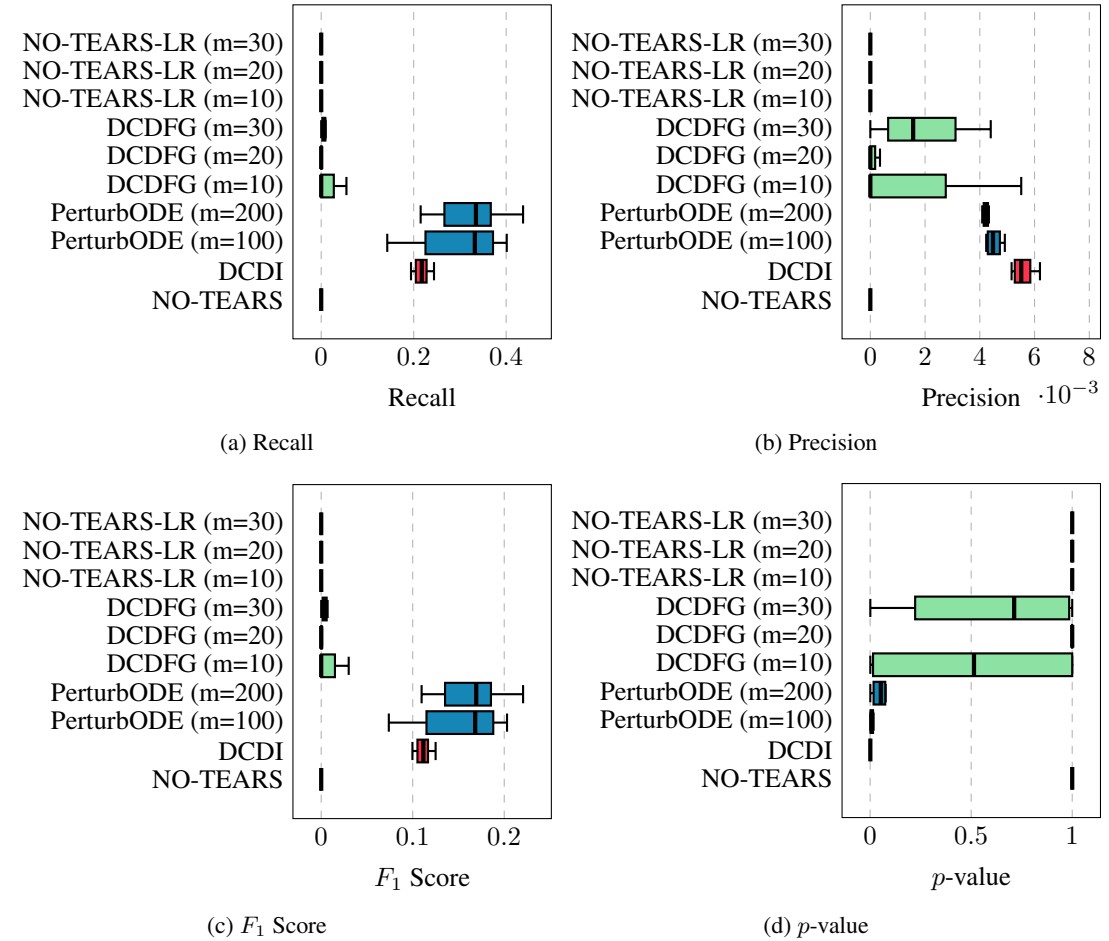

Figure 10: Perfect intervention over-expression (CRISPR-a) SERGIO simulation GRN inference. Ground truth GRN is a known yeast GRN (400 genes). Models with different number of modules are compared.

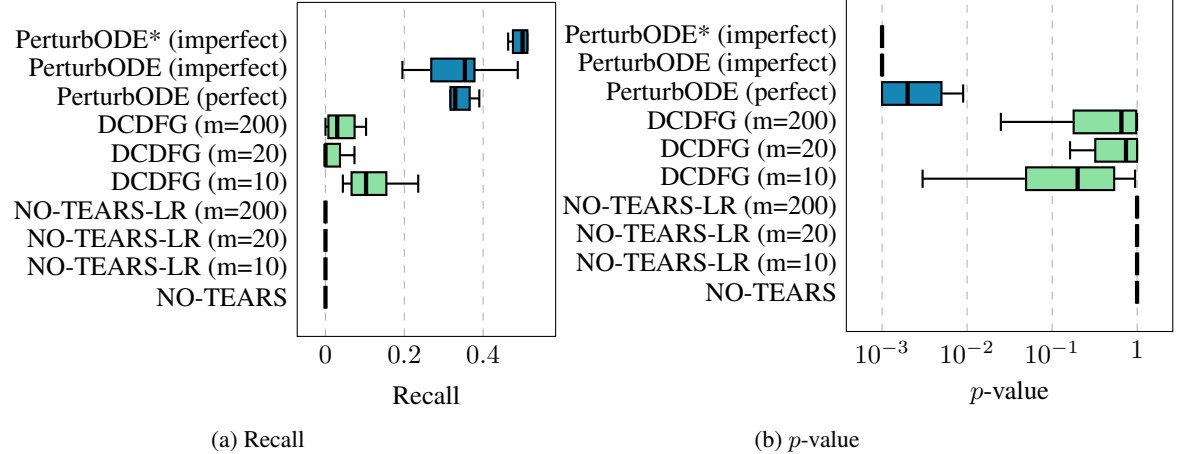

Figure 11: GRN Inference on TF Atlas Dataset (817 genes). Models with different number of modules are compared.

## 6.6 ABLATION STUDY

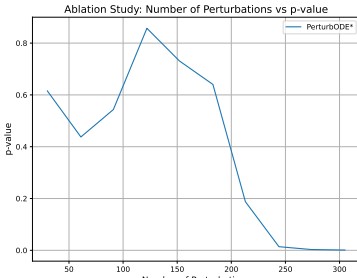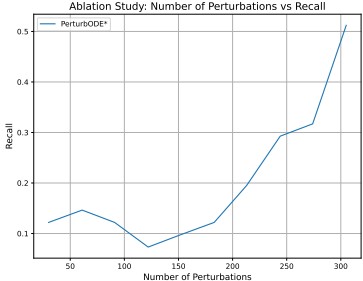

Figure 12: Ablation study: TF Atlas number of perturbations v.s. recall and p-value.

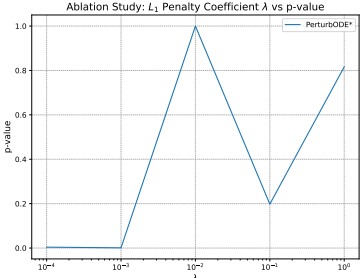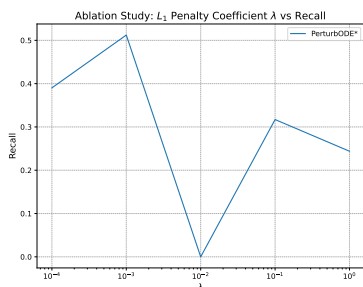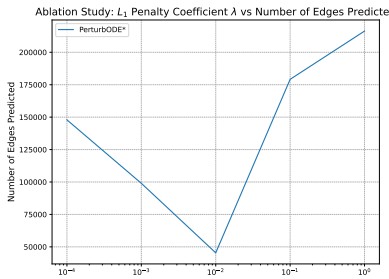

Figure 13: Ablation study: TF Atlas $L_1$ penalty coefficient $\lambda$ v.s. recall, p-value, and number of edges predicted.

Ablation study is done for PerturbODE* trained on TF Atlas. Figure 12 shows the number of perturbations included for training plotted against recall and p-value. It is clear that as the number of perturbations grow, recall increases and p-value decreases. Figure 13 shows the change in recall and p-value when varying the $L_1$ penalty coefficient for $B$. Ablation study shows that PerturbODE* yields statistically significant result when $\lambda \leq 0.001$. Further, it is evident that as $\lambda$ increases above 0.01, the number of edges predicted increase again. Our GRN is encoded as $\mathbf{W} = A \operatorname{diag}(\alpha \circ \mathbf{1}_N) B$. The multiplication of sparse matrices is not necessarily sparse. Further analysis shows strong penalization of $B$ leads to overly dense $A$, as the model resorts to $A$ for data fitting. This could lead to a rise of the number of edges predicted.

### 6.6.1 PREDICTION ON UNSEEN INTERVENTION ALL UMAP AND PCA PLOTS

Figures 14, 14, show the detailed results on prediction on test data (unseen intervention) through UMAP and PCA.

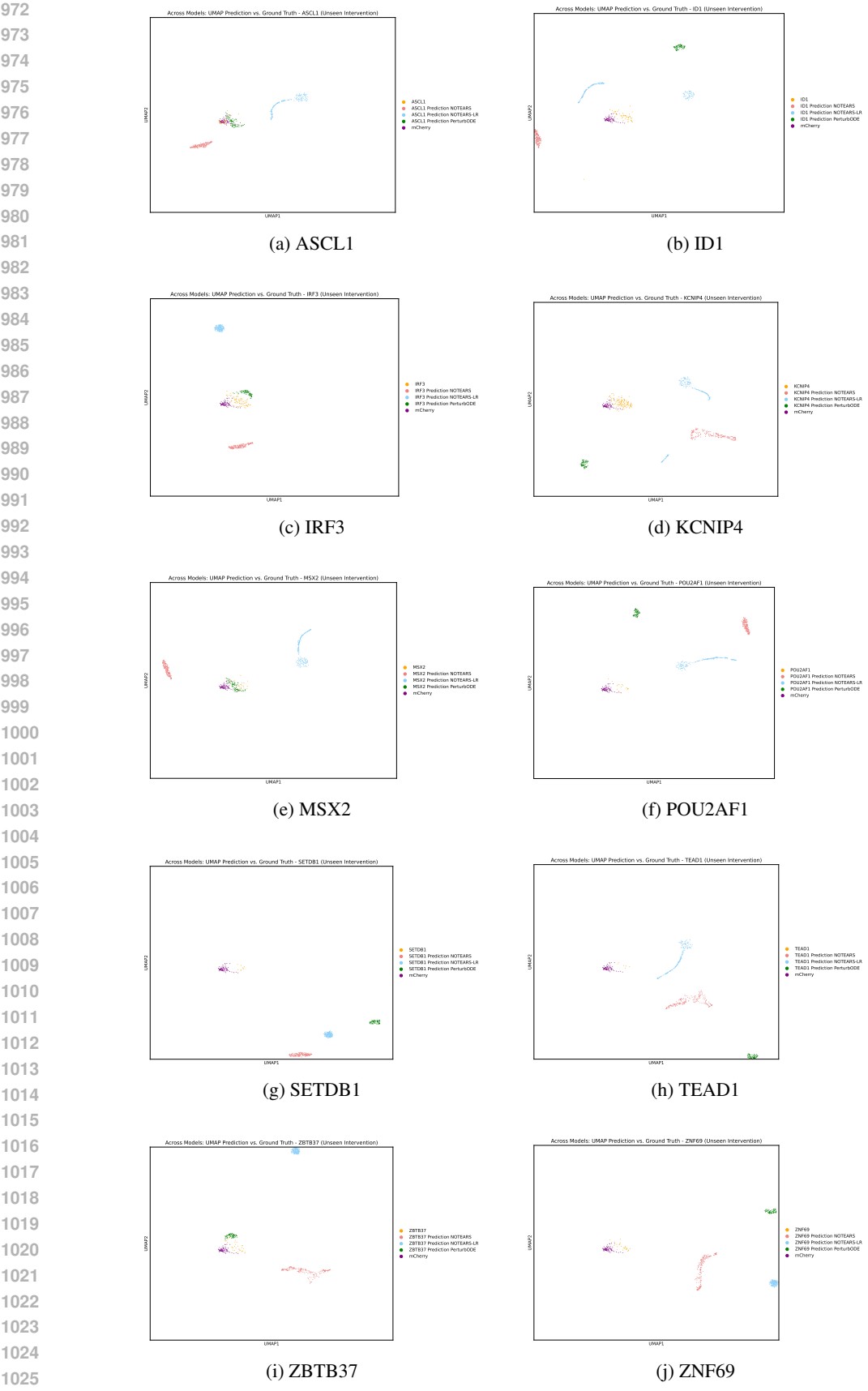

Figure 14: UMAP of predictions on unseen interventions across models.

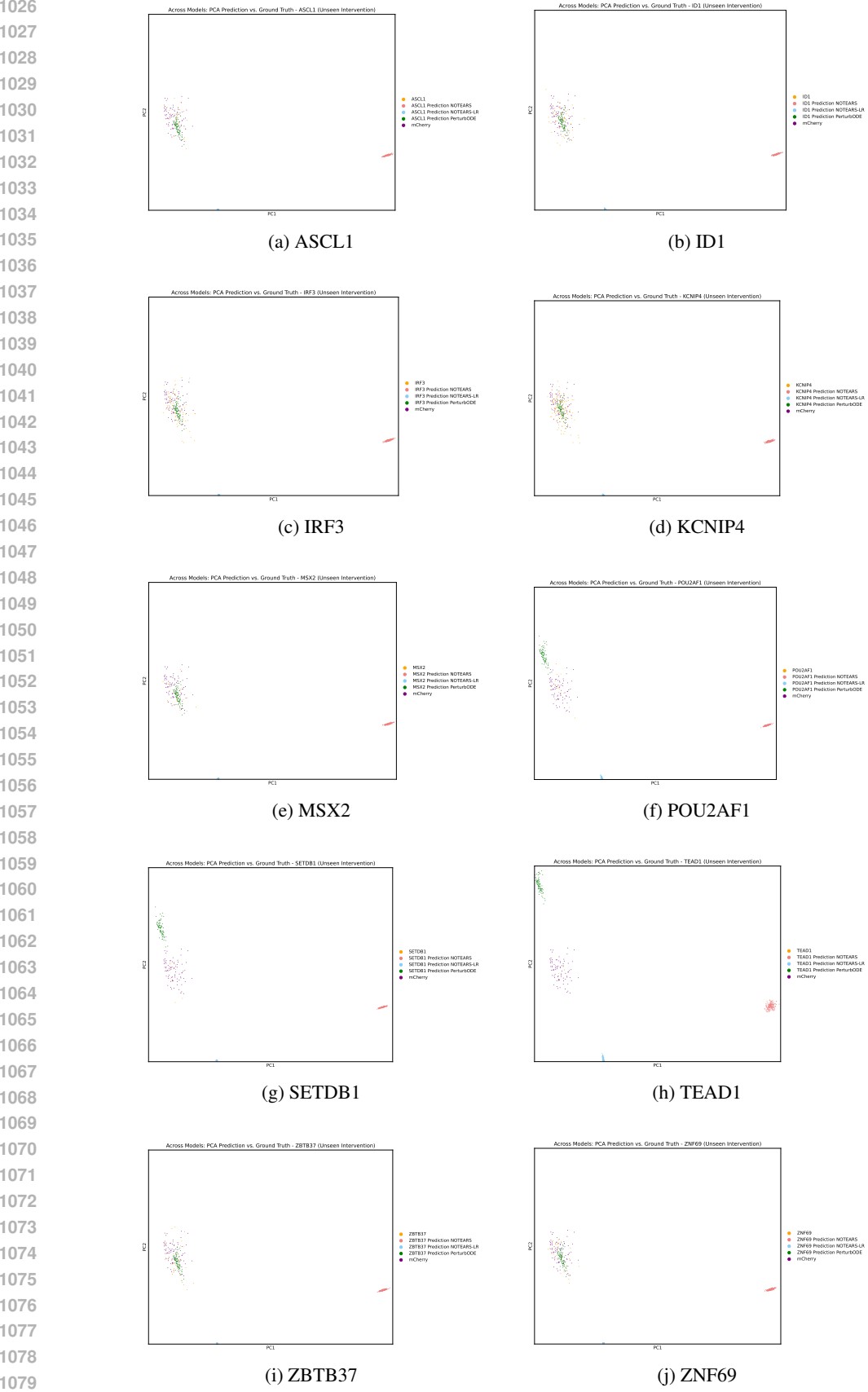

Figure 15: PCA of predictions on unseen interventions across models.

### 6.7 PERTURBODE MODEL TRAINING

After training, the average $W_2$ distance on both the training and held-out validation datasets decreases significantly and converges. The convergence rate of the $W_2$ distance varies for each TF in the training and validation sets.

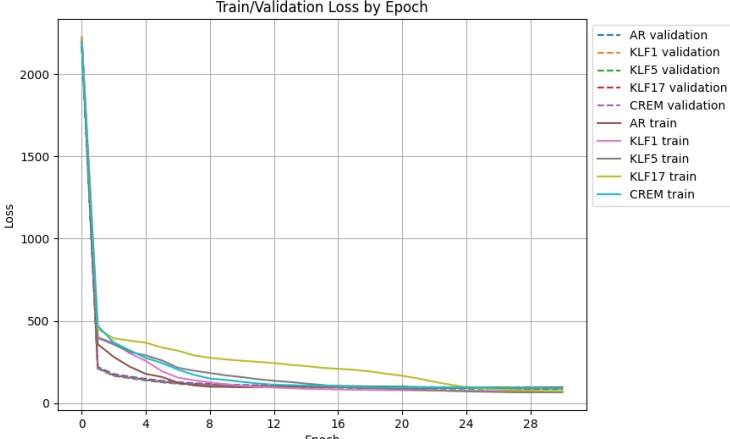

Figure 16: Convergence of $W_2$ losses for trajectory predictions of training and validation samples per TF. Average validation loss on TF Atlas is 78.88.

### 6.8 GROUND TRUTH GRNS FROM TF ATLAS

The three GRNs with high confidence inferred in Joung et al. (2023) are consistent with their induced cell types and roles in development. GRHL1 and GRHL3 target TFAP2C and the TEAD family of TFs to induce trophoblasts, while FLI1 targets AP-1 family TFs (such as JUN and FOS) and ETV2 to induce vascular endothelial cells (Krendl et al., 2017; Dejana et al., 2007). The GRN consisting of CDX1, CDX2, and HOXD11-influences posterior HOX genes is known to contribute to the definition of the anterior-posterior axis (Neijts et al., 2017). The three GRNs are in Figures 17, 18, 19.

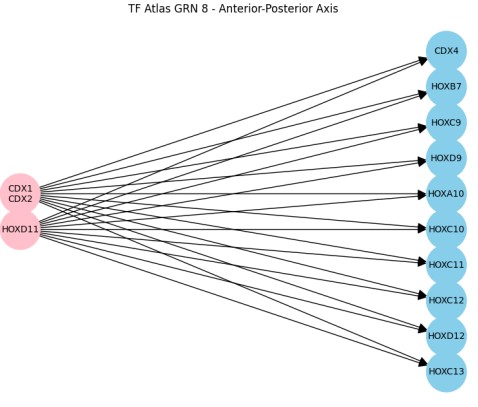

Figure 17: GRN with high confidence from TF Atlas - GRN8

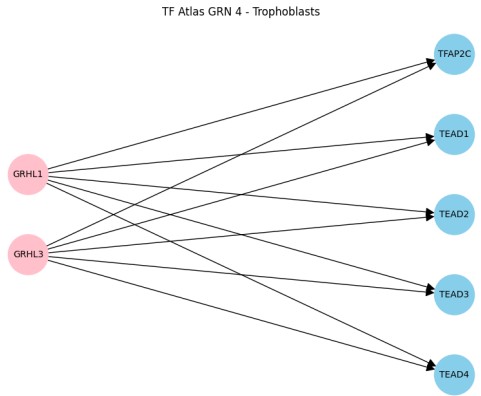

Figure 18: GRN with high confidence from TF Atlas - GRN4

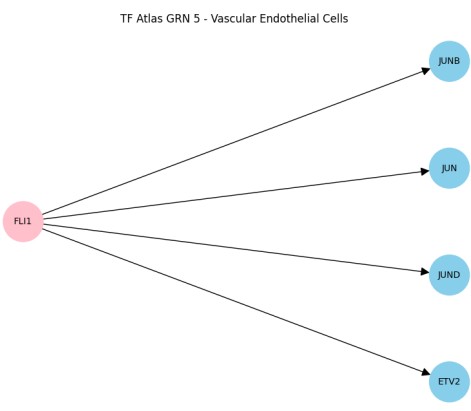

Figure 19: GRN with high confidence from TF Atlas - GRN5

## 6.9 SERGIO SIMULATION

SERGIO proposes simulation of scRNA-seq data by sampling a directed acyclic GRN through a SDE (Dibaeinia and Sinha, 2020). Although SERGIO does not support interventional data, we modified its framework to simulate gene over-expression with perfect interventions (CRISPR-a). For each interventional regime $I \in \mathcal{I}$, the SDE is parameterized in the following Equation 8.

$$dX_t = \left( M\left( P(X_t) - \lambda \circ X_t \right) + \sum_{j \in I} \gamma_j \cdot \delta_j \right) dt + q \circ \left( \sqrt{P(X_t)} dW_\alpha + \sqrt{\lambda X_t} dW_\beta \right) \quad (8)$$

The infinitesimal change of expression level (which is the stochastic process $X_t$) of gene $j$ at time $t$ over an infinitesimal time interval $dt$, denoted as $(dX_t)_j$, is governed by its production rate $P_j(X_t)$, which is modulated by its regulators according to a given GRN in Equation 9. It also depends on the decay rate $\lambda \in \mathbb{R}^d_+$ and the noise amplitude $q \in \mathbb{R}^d$ influencing its transcriptional variability. $M$ and $\sum_{j \in I} \gamma_j \cdot \delta_j$ are the masking matrix and the over-expression term analogous to those in Equations 1 and 2.

$$P_j(X) = \sum_{j=0}^{d} p_{ji}(X) + b_j \quad \text{for } p_{ji} \text{ in 10 , 11} \quad (9)$$

$$p_{ji}(X) = K_{ji} \frac{X_i}{h + X_i} \quad \text{if regulator } i \text{ is an activator of gene } j \tag{10}$$

$$p_{ji}(X) = K_{ji} \left( 1 - \frac{X_i}{h + X_i} \right) \quad \text{if regulator } i \text{ is a repressor of gene } j \tag{11}$$

For each pair of genes $i$ and $j$, the coefficients are initialized as in 12.

$$\lambda_j \sim \mathcal{N}(0.8, 0.2)_+ \quad , \quad K_{ji} \sim \mathcal{U}(0, 5) \quad , \quad q_j \sim \mathcal{U}(0.3, 1) \quad , \quad \gamma_j \sim \mathcal{N}(10, 1)_+ \quad ,$$

$$h = \frac{1}{d} \sum_{j=0}^{d} \frac{b_j}{q_j} \quad ,$$

$$b_j \sim \mathcal{N}(10, 0.01)_+ \quad \text{if gene } j \text{ is a master regulator},$$
$$b_j = 0 \quad \text{if gene } j \text{ is not a master regulator.} \tag{12}$$

$W_\alpha, W_\beta \in \mathbb{R}^d$ are two independent Wiener processes. We numerically simulate the SDE in Equation 8 using the Euler-Maruyama Scheme (E et al., 2019) with $\Delta t = 2$ in 50 steps.

$$(X_j)_{t+\Delta t} = (X_j)_t + \left( \left( P_j(X_t) - \lambda_j X_j(t) \right) \cdot \mathbb{I}_{j \notin I} + \gamma_j \cdot \mathbb{I}_{j \in I} \right) \Delta t$$
$$+ q_j \sqrt{P_j(X_t)} \Delta W_\alpha + q_j \sqrt{\lambda_i X_j(t)} \Delta W_\beta \tag{13}$$

$$(\Delta W_\alpha)_j \sim \sqrt{\Delta t} \mathcal{N}(0, 1), \quad (\Delta W_\beta)_j \sim \sqrt{\Delta t} \mathcal{N}(0, 1) \tag{14}$$

Lastly, the SDE 8 is initialized at the expected fixed point $X_0$ (where the drift of the SDE vanishes) with over-expression but without masking (perfect intervention). SERGIO assumes Jansen's Equality $E[p_{ji}(X_i)] \approx p_{ji}(E[X_i])$ for simplicity of initialization (Dibaeinia and Sinha, 2020). Hence, $X_0$ is initialized to the following expectations in Equations 15 and 16:

$$E[X_j] = \frac{\sum_{i=0}^{d} p_{ji}(E[X_i])}{\lambda_j} + \gamma_j \cdot \mathbb{I}_{j \in I} \quad \text{if } j \text{ is not a master regulator} \tag{15}$$

$$E[X_j] = \frac{b_i}{\lambda_j} + \gamma_j \cdot \mathbb{I}_{j \in I} \quad \text{if gene } j \text{ is a master regulator} \tag{16}$$

When simulating data using SERGIO, we use a real yeast GRN ($dim = 400$) and 10 random DAGs ($dim = 100$) with 500 binary entries (1 or 0). For clarity of comparison across models, the real yeast GRN is pruned to enforce acyclicity and include only positive directed edges. For both scenarios, the synthetic dataset generated by SERGIO includes 10,100 cells, created from 100 intervention schemes, each targeting 5 genes, along with one non-intervention scheme. Each regime provides 100 observations.

### 6.10 GENE MODULE EXAMPLE: FLAGELLA OF E. COLI

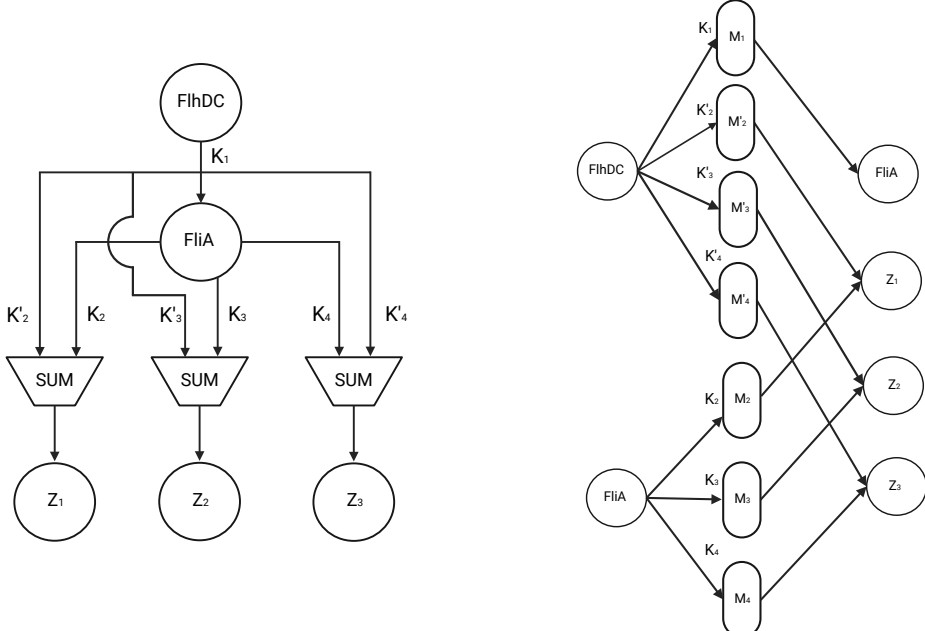

Figure 20: Regulatory circuit for the production of flagella in E. coli.

It is well established that the regulatory circuit responsible for the production of E. coli follows the network motif of multiple-output Feedforward Loop (Alon, 2006, pp. 64-68). Its circuit is shown on the left of Figure 20, where FlhDC and FliA regulate $Z_1$, $Z_2$, and $Z_3$, which are operons encoding the proteins that make up the flagella of E. coli. (In fact, there are in total 6 operons for this process.) Each operon consists of a group of genes, and it is regulated by a weighted sum of non-linearly activated signals from FlhDC and FliA through Hill functions.

The order in which the operons are activated matches the order of proteins needed to assemble the flagella. The timing of activation is achieved by different activation thresholds in the Hill functions. If $Z_1$ is activated before $Z_2$, which is activated before $Z_3$, then $K_2 < K_3 < K_4$. In other words, $Z_1$ needs a lower concentration of FliA to be switched on. For example, $Z_1$ would include the group of genes encoding the proteins for MS ring (base of flagella) and $Z_3$ would be for the filament (tail of flagella). In PerturbODE, the activation threshold is tuned by the bias term, $\beta$, to the hidden neurons.

This structure can be represented in a two-layer MLP shown on the right of Figure 20. Each operon $Z_i$ is regulated by the weighted sum of signals from two modules $M_i$ and $M_i'$. The signals from FliA and FlhDC are first activated by Hill functions with different activation thresholds before being transferred to modules $M_i$ and $M_i'$ respectively.

To represent this gene regulatory circuit with an adjacency matrix $\mathbf{W} = A \operatorname{diag}(\alpha \circ \mathbf{1}_N)B$, we multiply the two coefficient weight matrices of the MLP with an additional scaling to account for the rate of activation controlled by $\alpha$.

### 6.11 STATISTICAL INFERENCE: STABILITY ANALYSIS

We bootstrapped (sampled with replacement) TF Atlas dataset 27 times to evaluate consistency in the edges selected by PerturbODE*. We also filtered the list of TFs perturbations that PerturbODE* trains on down to the TFs pertinent to the ground truth GRNs. Then the gene expression space is the

union between the filtered TF list and the top 50 highly variable genes, resulting in 52 genes. Future versions will include experiment with more bootstrapped datasets.

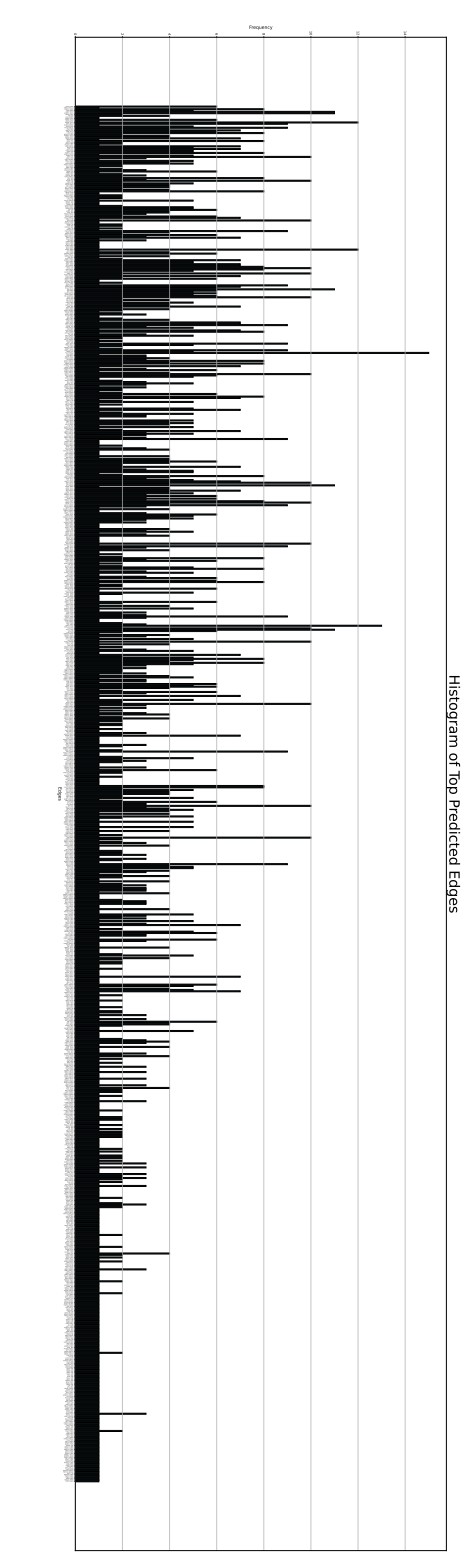

Figure 21: Histogram: Top Genes Selected by PerturbODE*.

Figure 21 shows the frequencies of top edges selected by PerturbODE*, which is trained separately on each bootstrapped dataset. We selected the top $100$ edges with the highest weights from each trained model. There is nontrivial variation in the top edges selected by PerturbODE. However, the top edges that are consistently selected agree with known TF interactions, such as GATA3–CDX1, HOXD11–SALL4, HOXD9–SALL4, and so on.

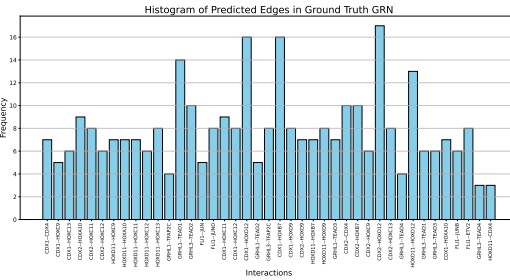

Figure 22: Histogram: Ground Truth Edges Selected by PerturbODE*.

Figure 22 illustrates the frequencies of ground truth GRN edges selected by PerturbODE* (threshold $c = 0.5$). There is also some variation in the edges selected, but the edge selections are overall consistent.

## 6.12 GENE ENRICHMENT ANALYSIS

We performed gene enrichment analysis using the Reactome Pathway Database (2022) and the Gene Ontology Biological Process (2021) with hypergeometric test. The examined pathways were filtered to those relevant to the anterior-posterior axis and vascular endothelial cells. The upstream genes and downstream genes of each module are selected by taking those edges whose weights are greater than 2 standard deviations of $B$ and $A$ respectively. Figure 25 illustrates the clustering of modules based on specific functions. A significant number of modules exhibit enrichment for anterior-posterior specification— a pathway crucial in development. This observation is expected, considering that the TF Atlas comprises human embryonic stem cells.

To show that the modules are not selecting identical genes, we plotted histograms of genes selected by various modules. Figure 23 shows a histogram of genes selected by the highlighted modules we selected for evaluation in Section 4.2.3, and Figure 24 showcases that of 10 randomly selected modules. Both histograms show clear clustering of gene selections by modules.

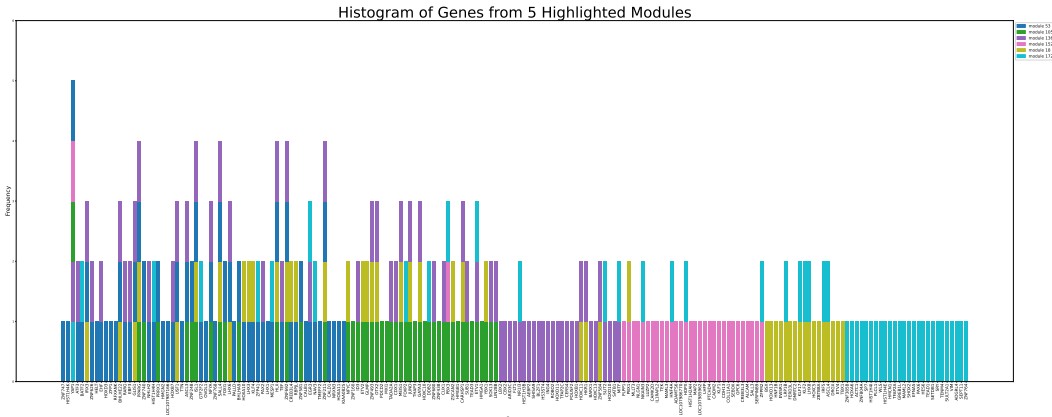

Figure 23: Histogram of Genes from 5 Highlighted Modules.

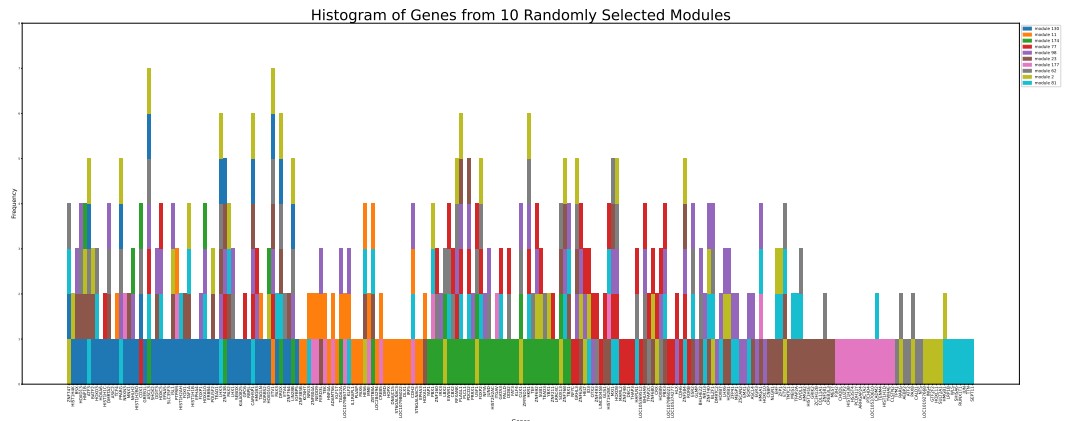

Figure 24: Histogram of Genes from 10 Randomly Selected Modules.

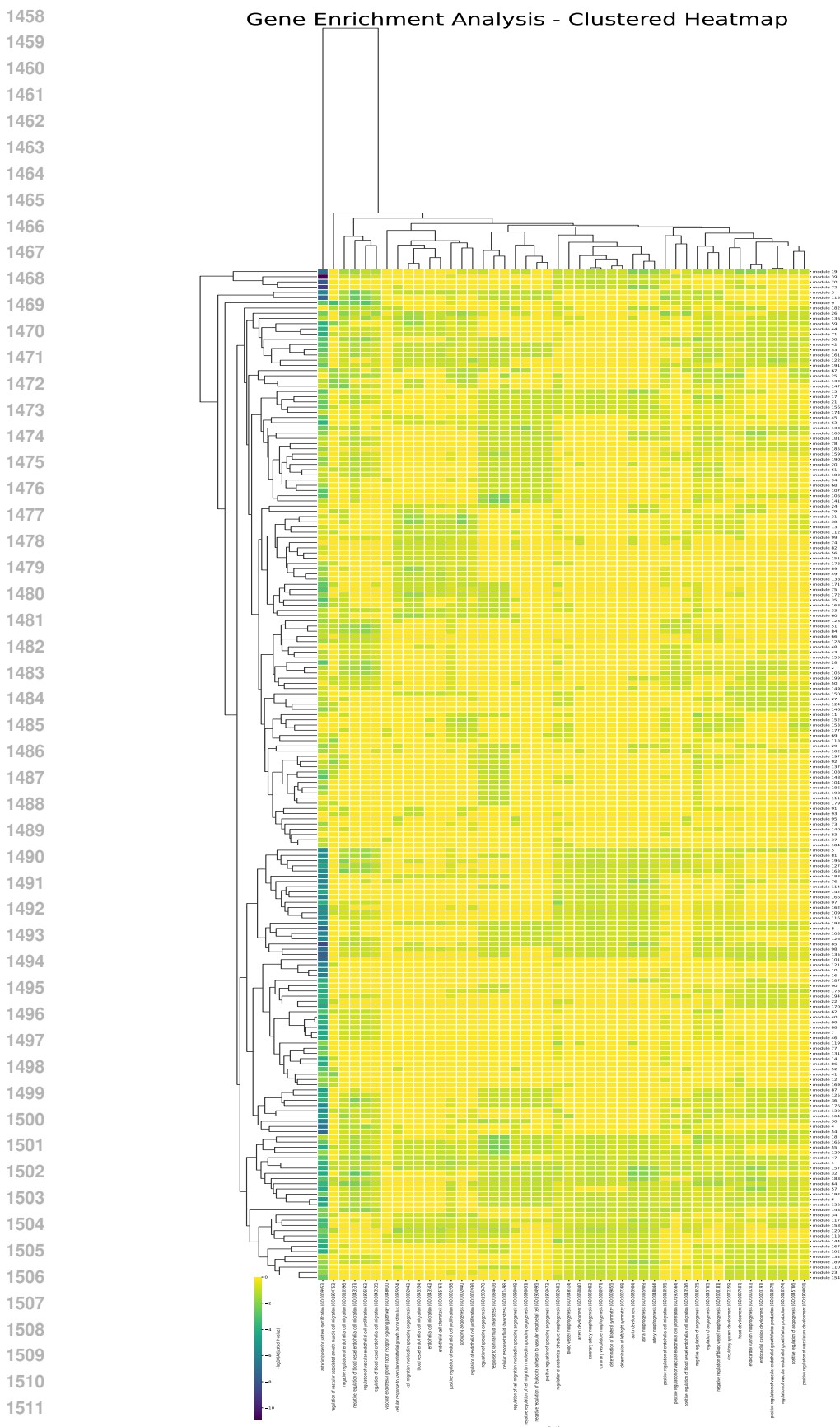

Figure 25: Gene enrichment clustered heatmap (average linkage) for all modules.

