# OpenReview forum: "Interpretable Neural ODEs for Gene Regulatory Network Discovery under Perturbations"
_ICLR.cc/2025/Conference — Submitted to ICLR 2025_

### Official Review · Reviewer_ZGrZ · 2024-10-24

**Soundness:** 2
**Presentation:** 3
**Contribution:** 2
**Rating:** 3
**Confidence:** 4

**Summary:**

The authors propose PerturbODE, a framework based on neural ODEs to infer causal GRNs from scRNA-seq data. PerturbODE aims to address limitations of prior methods regarding scalability, and modeling capacity. Two simulated experiments are performed as well as an experiment using real-life perturbation data (TF atlas).

**Strengths:**

- The study is well-motivated and the writing is clear. Addressing causality in GRN inference is important as most methods rely on gene co-expression networks.
- The proposed method scales well to large datasets, which is essential in an era where large-scale omics datasets are increasingly common.
- The model can handle both perfect and imperfect interventions, which can model a wide range of experimental conditions.

**Weaknesses:**

My main concerns with this paper relate to experiments. It is not evident from the validation analysis that the proposed method is superior. Furthermore, the study would benefit from incorporating a larger number of real life datasets and baselines. See detailed comments below.

- There is a large discrepancy in the number of selected edges between PerturbODE and the baselines for the atlas TF dataset (103,941 for PerturbODE and fewer than 500 for NO-TEARS and NO-TEARS-LR). It is not clear how meaningful the reported recall scores are given such discrepancy. Tuning parameters to select more edges in the baselines or including other baselines from causality literature [1-2] is necessary to provide a more accurate comparison.
- The total number of genes used in the TF atlas is 817, for a total of 667,489 possible edges. Selecting 100k from these (around 1/6) will likely inflate recall scores and, furthermore, is not very biologically plausible (you would expect around 3 interactions per gene on average [3]).
- Only one real experimental dataset is used (TF atlas) in the benchmark. Incorporating other datasets (e.g., from drug treatment, environmental stress) is necessary to support the claims made in this paper.
- The analysis is largely performed on edges and gene modules. More biological validation, e.g., gene enrichment analysis of top regulated genes, would enhance the biological validity of the model.
- Comparing against established methods for (non-causal) GRN inference (e.g., from the BEELINE benchmark [4]) would help the reader determine the utility of such causal models.
- Other methods have proposed neural ODEs for GRN inference [5]. A comparison with these or explanation why they are not applicable may be necessary.

[1] [https://arxiv.org/abs/2301.01849](https://arxiv.org/abs/2301.01849#)

[2] https://arxiv.org/abs/2107.10483

[3] https://academic.oup.com/nar/article/50/W1/W398/6591524

[4] https://www.nature.com/articles/s41592-019-0690-6

[5] https://genomebiology.biomedcentral.com/articles/10.1186/s13059-024-03264-0

**Questions:**

- Would help to show $-\log_{10}$(p-values) rather than p-values in plots.
- How important is the decay component in (1)? RNA decay may or may not be plausible for cells 7 days after perturbation.
- As I understand it, the TF atlas consists of 2 time points. Can the method work with scRNA-seq datasets with multiple time points?

---

> ### Author Response · Authors · 2024-11-28
>
> In response to weaknesses:
>
> * It is a good point that there are large discrepancies in the number of edges predicted. We have included the recall scores plotted against different sparsity levels for experiment on TF Atlas. (see appendix 6.53) In addition, I have now included AUPRC for the results for simulated data. In addition, I have added a statistical inference with stability analysis in appendix 6.11 for further evaluation of edge selection.
>
> * Large scale GRN discovery methods tend to overpredict false positives. PerturbODE is intended to be a method that identifies potential gene regulatory networks for biologists, who would conduct more in-depth experiments to manually validate the potential gene regulatory networks, preferably with Chip-seq data.
>
> * PerturbODE is designed to train on datasets with hundreds of perturbations with known interventional targets. Uncertainty could arise when discerning which genes are under perturbation given drug treatment and environmental stress. However, it would be interesting to see PerturbODE training on these datasets (drug treatment and environmental stress) with multiple time points in the future.
>
> * Thank you for your suggestion! I have now included a gene enrichment analysis in section 4.2.3 and more details in appendix 6.12. I also plotted histograms of genes selected by modules in appendix 6.12.
>
> * It is a great suggestion to compare to methods in BEELINE. It is a good future direction that will be included in later draft of this work.
>
> * Other ODE methods, such as [1] and [2] were mostly designed for time series data with a single perturbation or no perturbation. Further Phoenix processes data as aligned pseudotrajectories. It is a completely different task as PerturbODE deals with cells that are not aligned. In theory, we could infer pseudotime on TF Atlas and use pseudotrajectories or pseudotime as time series data, but this would require large variation to the current architecture. However, it would be interesting to compare PerturbODE on time series data for future evaluation.
>
> In response to questions:
>
> * log scaled p-values have been included in results.
>
> * Decay is linear, and it should always be present in the biological system. At equilibrium the decay is balanced out with basal and regulatory expressions. For instance, let W be a positive scalar representing decay, and b is a positive scalar for basal expression. X(t) is a dynamical system describing the evolution of gene expressions. dX/dt = b - WX. At equilibrium, b - WX* = 0. Hence, X* = b/W. Decay and basal expression still happen simultaneously, but they are cancelled out infinitesimally. I hope that I understood your question correctly. If not, feel free to make further comments. (Decay exists mostly to maintain stability. From an ODE perspective, it creates a trapping region that ensures a maximum gene expression level. Since the MLP part of the ODE is bounded, as X grows larger, the linear decay -WX will eventually dominate and decrease expression again. )
>
> * It would be interesting to include experiments on datasets with multiple time points in a later draft of this work. Thank you for your suggestion!
>
> [1] Hossain, I., Fanfani, V., Fischer, J., Quackenbush, J., & Burkholz, R. (2024). Biologically informed NeuralODEs for genome-wide regulatory dynamics. bioRxiv. https://doi.org/10.1101/2023.02.24.529835
>
>  [2] Jackson, C. A., Beheler-Amass, M., Tjärnberg, A., Suresh, I., Shang-mei Hickey, A., Bonneau, R., & Gresham, D. (2023). Simultaneous estimation of gene regulatory network structure and RNA kinetics from single cell gene expression. Center For Genomics and Systems Biology, New York University.

---

> ### Comment · Reviewer_ZGrZ · 2024-11-29
>
> I appreciate the efforts of the authors to address my concerns. I think the work has potential, but it needs to go through a round of revision before it is ready for publication. I maintain my current score and look forward to a future version that is more adequate. I hope the comments below will help improve the future version of this work.
>
> 1. I understand that PerturbODE performs better at different levels of sparsity. I think this would have to be on the main text. I really do not think that the recall scores reported on the main text are informative given the large discrepancy in recall scores.
> 2. Not sure if I understood the response to my question number 2. Doesn't the large number of selected genes also mean that your method will also overpredict false positives?
> 3. Drug datasets would indeed be interesting to see. You can focus on datasets where the targets of the drugs are known.
> 4. I thank the authors for adding gene enrichment analysis results. Doing a similar thing for the baselines and comparing them might help understand which method is selecting more biologically relevant genes.

---

### Official Review · Reviewer_kwHy · 2024-11-01

**Soundness:** 3
**Presentation:** 2
**Contribution:** 3
**Rating:** 5
**Confidence:** 3

**Summary:**

The paper titled “Interpretable Neural ODEs for Gene Regulatory Network Discovery under Perturbations” presents PerturbODE, a neural network designed to model cell state trajectories and derive causal gene regulatory networks (GRNs) from its parameters. The authors propose a model that utilizes neural networks to infer changes in gene expression levels based on their current states. A notable contribution of this work is its ability to model GRNs with negative edges and explicitly specify perturbed genes. The study demonstrates PerturbODE’s effectiveness in deriving the ground truth GRN and highlights its potential to predict cellular responses to novel perturbations.

**Strengths:**

1.PerturbODE offers excellent interpretability, with parameters that reveal clear relationships between genes and the inferred gene modules.

2.The innovative approach of PerturbODE effectively captures key biological processes, such as cellular differentiation and negative feedback regulation.

3.The authors have conducted comprehensive experiments, comparing PerturbODE with other methods on both simulated and large-scale perturbational scRNA-seq datasets. They also explore its capability to identify negative autoregulation and infer gene modules.
4.The paper is well-organized, featuring clear explanations of the methodology.

**Weaknesses:**

1.It would be beneficial to include more discussions and experiments addressing instances where PerturbODE does not outperform other models.

2.In Section 4.2.1, a more detailed discussion about performance variations could enhance clarity and understanding.

**Questions:**

1.Have you considered using a more complex or simpler model? How might this impact performance?

2.Could you elaborate on how this work contributes to our understanding and discovery of gene modules?

3. In the experimental phase, the paper primarily presents metric-based results, showing that the method outperforms others. However, in solving a specific GRN problem, the focus should be on whether the inferred GRN can identify key genes or transcription factors (TFs) that can then undergo downstream analysis. For a biological application, simply comparing metrics does not sufficiently demonstrate the model’s effectiveness.

4. In GRN inference problems, perturbation experiments for some key genes are also an important downstream analysis. Including some perturbation experiments could help validate the accuracy of the inferred GRN.

---

> ### Author Response · Authors · 2024-11-28
>
> In response to weaknesses:
>
> 1. PerturbODE doesn't outperform DCDI, but DCDI suffers in scalability. Therefore, DCDFG was developed with additional low-rank structure to improve scalability. I have now given more discussions in section 4.1. Further, it would be interesting to include more experiments with cyclic networks simulated by BoolODE in future draft of this work as suggested by other reviewer.
>
> 2. More discussions about performance variations have been added in section 4.1.
>
> In response to questions:
>
> 1. Thank you for your question. Model complexity could be varied by changing the number of modules in the hyperparameter. See appendix 6.2.2 for the change in validation loss with varying number of modules used.
>
> 2. PerturbODE's main contribution is a highly scalable and biologically realistic approach to discover gene regulatory networks at large scale from perturbation data.  Gene modules are discovered directly from model parameters, which allows efficient computation as well as adherence to biological realism. The gene module structure embedded in a two-layer MLP is inspired by the real network motif of multiple output feedforward loops known to be prevalent in yeast and E. coli. (see Appendix 6.10)
>
> 3. PerturbODE is intended to be a method that identifies potential gene regulatory networks for biologists, who would conduct more experiments to validate the potential gene regulatory networks with additional experimental Chip-seq data.  We argue the model's capacity to generalize to unseen perturbations does demonstrate its usefulness for application, as the modules it's learning are capturing something real about the biology.
>
> 4. Ideally, we would conduct perturbation experiments. It could be an interesting future direction with the right resources.

---

> > ### Comment · Reviewer_kwHy · 2024-11-29
> >
> > We are very much looking forward to future perturbation experiments mentioned in your paper. In fact, metrics alone often fail to capture the full picture. For example, in the field of computer vision, even if accuracy reaches 99%, it may still lack practical value. This is even more pronounced in biology, where metrics are a very weak form of evaluation. Only real-world applications that prove effective can truly demonstrate a model’s performance. Therefore, we will maintain our current score.

---

### Official Review · Reviewer_HJyF · 2024-11-04

**Soundness:** 2
**Presentation:** 3
**Contribution:** 3
**Rating:** 5
**Confidence:** 4

**Summary:**

The authors present PerturbODE, a method based on biologically informed neural ordinary differential equations for modelling perturbation-specific cell trajectories and inferring gene regulatory networks (GRNs) from perturbational (interventional) data. They empirically validate their approach through an extensive set of experiments on both synthetic and real systems. Through their empirical experiments, the authors show that PerturbODE can achieve state-of-the-art performance in high-dimensional settings for the task of GRN inference, relative to counterpart baseline methods.

**Strengths:**

This work addresses the challenging problem of GRN inference and cell trajectory inference - both important problems in computational biology. The authors propose a novel method based on neural ODEs that incorporates biologically informed dynamics to address both problems. PerturbODE has three key strengths that work towards addressing these longstanding problems in the computational biology community:

1. GRN inference over high-dimensional systems is a challenging task. The authors demonstrate that their novel method, PerturbODE, is able to learn GRNs for high-dimensional systems.
2. A challenging problem in trajectory inference of cells is predicting the response of cellular systems under unseen interventions/perturbations. The authors show that PerturbODE is able to achieve improved performance on this task relative to counterpart baselines.
3. PerturbODE is able to model and learn cyclic dependencies, pertinent to GRN inference.

**Weaknesses:**

Although this paper works towards addressing important and challenging problems of GRN inference and cell response prediction, there remain several shortcomings that limit the contributions of this paper (see below).

- If one of the claims is that PerturbODE can model cyclic dependencies between variables (genes), why evaluate on systems where the data generative processes adhere from directed acyclic graphs (DAGs)? If I understand correctly, SERGIO is limited to simulating systems given a DAG representation of a GRN. In this case, it may be useful to consider other biological system simulation tools in addition to SERGIO, for example, the framework in [1] could be used. At the minimum, this can be done in smaller systems (10-100 genes) to empirically validate the claims that PerturbODE can model cyclic dependencies.
- Likewise, for results on GRN inference on the transcription factor (TF) atlas, it is not clear if the ground truth GRNs are DAGs or contain cycles. In 4.2.2, the authors discuss that PerturbODE is able to predict negative feedback loops. However, in Appendix 6.7, the ground truth GRNs which are used for evaluation on the TF atlas (shown in Figures 15, 16, and 17), appear to be acyclic. It is not clear how the assessment of PerturbODE's ability to learn cyclic dependencies in GRNs is accomplished in this section. Further explanation and details are required to convey the significance of this result (and likewise this contribution).
- This work focuses on the task of GRN inference from interventional data. However, only a select few baseline methods are considered and compared. There exist many methods specifically tailored for GRN inference. To give a few examples, I refer the authors to [1]. I think it is okay if other GRN inference methods are not included, but it would be beneficial to include some discussion and justification on why the baselines considered in this work are sufficient for fair evaluation.
- For experiments in 4.2.1 for predicting response on held-out interventions. The authors compare only to baselines that are not necessarily tailored for this task. There exist state-of-the-art methods that address this problem, for example [2, 3]. To effectively showcase the utility of PerturbODE, it would be beneficial to compare to an existing method designed for predicting response to perturbations in biological systems since the NOTEARS baselines are not designed for this task. Without a biological-specific baseline for predicting response to perturbations, it is difficult to assess how PerturbODE performs relative to state-of-the-art methods. Moreover, it would be helpful to include a baseline that does not learn a GRN and is just trained to predict the perturbational response. Such a method should not work in the setting of unseen interventions since it will never see those conditions, but can be informative of whether or not learning the GRN is actually beneficial for this task.

In general, there are still various details missing regarding experimental details, empirical validation of discovered GRNs, and justification of design choices of the method (see questions below). This makes the paper somewhat incomplete and hinders the presentation of the method and results. I think if these items are addressed, it would improve the overall quality of the paper.

**Questions:**

- Lines 245 - 246: Beyond use for benchmarking, what is the justification of classifying negative edges as non-existing edges? Why not take the absolute value over $\mathbf{W}$ and consider all edges as positive edges?
- The authors state that the structural hamming distance (SHD) metric would strongly favour predictions of empty graphs due to sparsity of ground truth GRNs (Lines 318 - 320). Do the methods tend to return empty graphs such that this would be the case? If the methods do return empty graphs, would this not be reflected in the other metrics, while still getting a performance metric through SHD that gives some global view on how the methods perform on predicting the GRN graph.
- For the experiments in 4.1 and 4.2, why not compute other additional metrics? For instance, given that the data has intervention, the negative log-likelihood of data points under intervention (I-NLL, done in DCDFG) could be computed. Possibly area under the ROC (AUROC) could also be computed to give an additional view of the results.
- In Figures 2 and 4, what are the mean and std for box plots computed over? Different graphs?
- For performance evaluation, the authors threshold $\mathbf{W}$ to construct a binary adjacency matrix so that the evaluation metrics can be computed. This is done by selecting some small value $\epsilon$ and setting edge weights below $\epsilon$ to $0$ and above $\epsilon$ to $1$. The determination of $\epsilon$ depends on a parameter $c$. This seems like a limiting factor in the sense that as $c$ is changed, performance may change
    - How do you decide the value of $c$? Is this value determined via SERGIO?
    - Is this treated as a settable parameter? If so, how would the results change as the value of $c$ is changed?
- On the SERGIO simulated data the authors claim that PeturbODE gives significantly higher precision, recall, and F1 scores compared to DCDFG, NOTEARS, and NOTEARS-LR, while comparable scores DCDI, especially in the high dimensional case (400 genes). Compared to DCDI, it appears PerturbODE also yields a significantly higher variance on Recall and F1 score, while performing overall worse on precision. Is there intuition as to why PerturbODE yields such a high variance on Recall? I assume since there is variance on recall, this leads to variance on F1 score as well.
- For the task of predicting the response of unseen interventions (In Table 1) is there intuition as to why PerturbODE performs worse on the 2-Wasserstein metric, but significantly better on Pearson correlation? In general, this is a very difficult task. It could help to observe how the numbers for these metrics look for seen interventions (see earlier question regarding Interventional-NLL).
- For the result presented In Figure 4, the authors state GRN inference is done for a system of 817 genes. But the ground truth GRNs shown in Appendix 6.7 have far less genes. Are the GRNs shown in Appendix 6.7 Figures 15, 16, and 17 only the non-zero edges in the ground truth GRNs?
- If the objective is to learn a flow map for pushing initial samples over time (eq 3), why not use the recent advances in flow matching [4], which can achieve the same result in a significantly more efficient manner using simulation-free training, compared to Neural ODEs? This could potentially enable further scaling.

References:

[1] Pratapa, Aditya, et al. "Benchmarking algorithms for gene regulatory network inference from single-cell transcriptomic data." Nature methods 17.2 (2020): 147-154.

[2] Lotfollahi, Mohammad, et al. "Predicting cellular responses to complex perturbations in high‐throughput screens." Molecular systems biology 19.6 (2023): e11517.

[3] Roohani, Yusuf, Kexin Huang, and Jure Leskovec. "Predicting transcriptional outcomes of novel multigene perturbations with GEARS." Nature Biotechnology 42.6 (2024): 927-935.

[4] Lipman, Yaron, et al. "Flow Matching for Generative Modeling." International Conference on Learning Representations. (2023)

---

> ### Author Response · Authors · 2024-11-28
> **In response to weaknesses:**
>
> In response to weaknesses:
>
> * We were previously not aware of BoolODE GRN models used in Beeline. We hope to include experiments measuring against that simulator in a later draft of the work.
>
> * Sorry about the confusion, we only suggest that PerturbODE resembles real gene regulatory systems through the adaptation of  the network motif (pattern) of negative autoregulation without any priors. Real gene regulatory systems in E. coli and yeast utilize negative autoregulation to stabilize gene expression. However, it would be great to simulate data with ground truth networks with cycles using BoolODE.   [1, pp 24-30] .
>
> * We agree there are many GRN inference methods and it's difficult to compare against all of them.  In this work we focused on comparing PerturbODE against other causal methods that directly encode GRN as matrices in neural networks and directly as matrices in linear models.
>
> * It would be interesting to include comparison to methods such as GENIE3 for further analysis. Other ODE methods, such as Phoenix and [2] were mostly designed for time series data with a single perturbation or no perturbation. Further Phoenix processes data as aligned pseudotrajectories. It is a completely different task as PerturbODE deals with cells that are not aligned. In theory, we could infer pseudotime on TF Atlas and use pseudotrajectories or pseudotime as time series data, but this requires significant variation to the current architecture. It would be interesting to experiment PerturbODE on time series data for future evaluation. For trajectory inference, Prescient runs on PCA space, while PerturbODE is designed to run on the direct gene expression space. Other methods such as scGEN are foundation models. It would not be a fair comparison since PerturbODE only trained on TF Atlas, but it would be interesting to experiment.
>
> [1] Uri Alon. An Introduction to Systems Biology: Design Principles of Biological Circuits. CRC Press Taylor & Francis Group, A Chapman & Hall Book, 2006.
>
> [2] Jackson, C. A., Beheler-Amass, M., Tjärnberg, A., Suresh, I., Shang-mei Hickey, A., Bonneau, R., & Gresham, D. (2023). Simultaneous estimation of gene regulatory network structure and RNA kinetics from single cell gene expression. Center For Genomics and Systems Biology, New York University.
>
>
> Response to questions is posted in a separate comment due to word limit.

---

> ### Author Response · Authors · 2024-11-28
> **In response to questions:**
>
> In response to questions:
>
> * While other models infer conditional densities for each edge, PerturbODE directly infers positive, negative regulation, or no edge. Evaluating edge existence using absolute values might benefit PerturbODE; however, I believe it's incorrect to label an edge as a true positive when the predicted edge is negative but the corresponding ground truth edge is positive.
>
> * For SHD, consider a sparse empty graph with 400 edges with dimension: 100x100. Method 1 predicts 300 of the 500 edges correctly but contains another 300 false positives. SHD for Method 1 would be 500, but SHD for an empty graph is 400, which is lower. On the other hand, precision/recall for Method 1 would be 0.5 and 0.6. Precision/recall for an empty graph would be 0 and 0. It would, however, make sense to use SHD if the ground truth network is dense.
>
> * For negative log likelihood, DCDFG learns conditional densities on each edge with predefined parameterization for the densities. For evaluation, unseen data is given directly to the model to compute the negative log likelihood (product of all conditional densities of edges). The same method would be impossible for PerturbODE since there is no predefined density. Additionally, I have now included AUPRC in the results.
>
> * I have now added the mean and std in a table in the appendix 6.5.1.
>
> * It is indeed a limiting factor that thresholds affect results. Therefore, we now have included AUPRC in results. Also, c isn’t a hyperparameter. c determines the threshold. It is chosen after training so that the model selects a reasonable number of edges (no more than 30%). (DCDFG uses a binary search to find the optimal threshold for the largest possible DAG.)
>
> * There is considerable variation in recall scores for PerturbODE especially in the simulated yeast dataset. This is likely due to the high sparsity in the ground truth GRN, which leads to weak signals in the simulated dataset. This results in false negatives. Further, L1 penalty is enforced on the individual matrix. As multiplication of sparse matrices is not always sparse, the number of predicted edges tends to fluctuate. Denser predictions would have higher recall scores.
>
> * For TF perturbations, when PerturbODE doesn't predict well it tends to shoot far off. However, in terms of the median W2 distance PerturbODE outperforms other models, and it does better in general. (see appendix 6.52)
>
> * The ground truth GRNs shown in appendix 6.7 are the known edges with high confidence. There is a limited number of well understood human GRNs as ground truths. It is an inevitable limitation in the field.
>
> * Flow matching isn't concerned with the "true" vector field; the paper focuses on a conditional linear vector field that induces the correct final distribution. But we're interested in the path a cell "actually" takes (which is very unlikely to be linear in expression space) and make biologically plausible assumptions via our parameterization of the ODE. For example, the decay term might not be learned in a flow matching objective, but is imperative for a biological model to guarantee cells reach a stationary state.

---

> > ### Comment · Reviewer_HJyF · 2024-11-28
> >
> > Thank you for answering my many questions and implementing some of my suggestions!
> >
> > In general, I am happy to see the authors implemented some of my feedback and improved the manuscript. With this, I am happy to raise my score from 3 -> 5. Unfortunately, I remain of the opinion that this work still requires further refinement, further justification of choices pertaining to the methodology, and further comparison with existing methods in the area.

---

### Official Review · Reviewer_Js4p · 2024-11-04

**Soundness:** 2
**Presentation:** 2
**Contribution:** 2
**Rating:** 5
**Confidence:** 4

**Summary:**

This paper introduces PerturbODE, a novel framework that uses neural ordinary differential equations (ODEs) to model gene regulatory networks (GRNs) from single-cell perturbation data. PerturbODE addresses limitations of existing causal graph discovery methods by (i) incorporating biologically-informed ODEs to model cell state trajectories under perturbations, (ii) does estimation in a latent space and (iii) derive GRNs from the ODE parameters by aligning latent and input space. The method can capture non-linear and cyclic gene interactions, which is an advantage over many existing methods and maps cell states to a lower-dimensional "gene module" space, which provides high-level insights into the system. Experiments on simulated and real single-cell RNA-seq datasets show PerturbODE outperforms existing methods in GRN inference and prediction tasks. The authors demonstrate PerturbODE's ability to recover biologically meaningful network structures like negative autoregulation.

**Strengths:**

- allows to model cyclic GRNs, which is not addressed that much in the literature yet. The authors also provide references that this is an important aspect of biological systems
- the model is quite intuitive in its formulation

**Weaknesses:**

- It's very hard to read Figure 5
- I am wondering how identifiable the model is and what number of perturbations is sufficient for this.
- there is no extension towards entropy-regularized OT
- an ablation study on the gamma parameter (scaling the sparsity effect) would be interesting to see

Typos:
- sometimes it's "non-linear" (l53) and sometimes it's "nonlinear" (e.g. l45)

**Questions:**

- It seems a subset of the modules is recovering relevant biological pathways. However, it only seems to be a very small amount? Can you comment on the other 95% of the modules? Does a Gene-set-enrichtment-analysis confirm your findings?
- Setting delta_j to a fixed value, here 1, would imply that all perturbations have the same strength on its gene. I guess this is not a reasonable assumption, since some perturbations won't have an effect at all?
- Can you clarify how you combine A and B exactly to get the GRNs?

I'm happy to adjust my score based on the answers to the Questions & Weaknesses subsections.

---

> ### Author Response · Authors · 2024-11-28
>
> In response to weaknesses:
>
> * Figure 5 has been updated.
>
> * Identifiability is a good question; although our empirical results suggest this model is identifiable as it can generalize to unseen perturbations, there are few proven results about identifying an ODE from interventional data, especially non-linear ODEs.  The paper [1] shows some barriers to proving results even for simple systems, and suggests the practical tactic of regularizing weight parameters, which we follow in our model.
>
> * Could you please elaborate on the suggestion about extension towards entropy-regularized OT?
>
> * It is a great suggestion to include an ablation study. I have included an ablation study in Appendix 6.6. Evaluating recall and p-value for PerturbODE trained on TF Atlas against various L1 penalty coefficient. Additionally, I evaluated recall and p-value with a various number of perturbations included in training.
>
> * Typo is fixed.
>
> In response to questions:
>
> * Regarding the other 95% of the modules, I have now included a gene enrichment analysis in section 4.2.3 for the 6 highlighted modules and an analysis for all modules in appendix 6.12. Thank you for your suggestion. Most modules show some enrichment for some pathways. I also plotted a histogram of genes selected by 10 randomly selected modules in appendix 6.12.
>
> * It is indeed more realistic to have different perturbation strength for each gene. I have included a version of our model with tunable perturbation strength for each gene, denoted PerturbODE*.  It outperforms other versions of our model in terms of GRN inference. Thank you for your input! Nevertheless, for prediction of response to unseen interventions, we used the original formulation with a fixed perturbation strength since it would be difficult to determine a perturbation strength for unseen interventions.
>
> * The GRN is essentially obtained by the multiplication of two coefficient weight matrices of the MLP with an additional scaling to account for the rate of activation controlled by α. The intuition is now added in Appendix 6.10 with a real example of E. coli flagella.
>
>
>
> [1] Aliee, Hananeh, Fabian J. Theis, and Niki Kilbertus. "Beyond predictions in neural ODEs: identification and interventions." arXiv preprint arXiv:2106.12430 (2021).

---

### Meta-Review · Area_Chair_XU1c · 2024-12-20

**Metareview:**

The paper introduces PerturbODE, a framework using neural ODEs to model cell state trajectories and infer causal GRNs, addressing limitations like scalability, linearity, and acyclicity in existing methods. Reviewers praised its novelty and biological relevance but raised concerns about validation, benchmarking, and biological interpretability, particularly regarding the large number of predicted edges and limited evidence for modeling cyclic dependencies. Despite improvements in the rebuttal, further validation and broader evaluations are needed.

**Additional Comments On Reviewer Discussion:**

Most reviewers engaged in the discussion and acknowledged the rebuttal. While they recognize the potential of the work, they believe it requires further revisions to reach a publishable standard.

---

### Decision · Program_Chairs · 2025-01-22

Reject